# Interannual evolutions of (sub)mesoscale dynamics in the Bay of Biscay

Guillaume Charria[1], Sébastien Theetten[1], Frédéric Vandermeirsch[1], Özge Yelekçi[1] and Nicole Audiffren[2]

[1]Ifremer, Univ. Brest, CNRS, IRD, Laboratoire d'Océanographie Physique et Spatiale (LOPS), IUEM, F-29280, Brest, France
[2]CINES, Monptellier, F-34090, France

*Correspondence to*: Guillaume Charria[1] (guillaume.charria@ifremer.fr)

**Abstract.**

In the North-East Atlantic Ocean, the Bay of Biscay is an intersection between a coastal constrained dynamics (wide continental shelf and shelf break regions) and an eastern boundary circulation system. In this framework, the eddy kinetic energy is one order of magnitude lower than in western boundary systems. To explore this coastal complex system, a high resolution (1 km, 100 vertical sigma layers) model experiment including tidal dynamics over a period of 10 years (2001-2010) has been implemented. The ability of the numerical environment to reproduce main patterns over interannual scales is demonstrated. Based on this experiment, the features of the (sub)mesoscale processes are described in the deep part of the region (*i.e.* abyssal plain and continental slope). A system with the development of mixed layer instabilities at the end of winter is highlighted. Beyond confirming an observed behaviour of seasonal (sub)mesoscale activity in other regions, the simulated period allows exploring the interannual variability of these structures. A relationship between winter maximum of mixed layer depth and the intensity of (sub)mesoscale related activity (vertical velocity, relative vorticity) is revealed and can be explained by large scale atmospheric forcings (e.g. the cold winter in 2005). The first submesoscale-permitting exploration of this 3D coastal system shows the importance of (sub)mesoscale activity in this region and its evolution implying a potentially significant impact on vertical and horizontal mixing.

**Keywords** : Bay of Biscay, (sub)mesoscale, mixed layer instabilities, interannual evolutions, coastal high resolution modelling

## 1 Introduction

As a semi-enclosed region, the Bay of Biscay (Figure 1) can be divided in three dynamical regimes: the circulation over the continental shelf, the transition region above the shelf break and the open ocean part. Our understanding of the general associated circulation in the Bay of Biscay has been progressively refined following the available observations and the improvement of numerical models. The first review by Koutsikopoulos and Le Cann (1996) introduced the general circulation patterns with a poleward circulation over the continental shelf, a poleward slope current and a general anticyclonic circulation in the open ocean. This general scheme has been detailed with new datasets from drifters in van Aken (2002), Charria et al. (2013), and Porter et al. (2016); from ADCP moorings in Batifoulier et al. (2012), Le Boyer et al. (2013), and Kersalé et al. (2015); from satellite altimetry in Herbert et al. (2011), and Le Henaff et al. (2011). Finally, most intense circulation patterns are today explained by intermittent coastal density driven jets disturbed by tidal dynamics over the continental shelf, a slope current with seasonal and interannual reversals meandering to generate eddies, and an open ocean region with a weak average circulation but several eddies propagating. From this statement, next questions to be

addressed to fulfil the scheme explaining the evolution of this coastal system concern the mesoscale and submesoscale dynamics.

The underlying mechanisms of the mesoscale and submesoscale activity in the ocean have been widely described and discussed during the past years (*e.g.* McWilliams, 1985; Capet et al., 2008a, b, c; Klein et al., 2008; Ferrari, 2011; Scherbina et al., 2013; Sasaki et al., 2014; Callies et al., 2015; Molemaker et al., 2015). This dynamics is particularly active (in terms of eddy kinetic energy) in western boundary currents.

In the present study, we aim at contributing to the description and the understanding of small scales features in eastern boundary regions where the average level of kinetic energy remains low (Caballero et al., 2008; Dussurget et al., 2011), and of the mesoscale and submesoscale activity impact on long-term fluctuations related to evolutions in atmospheric conditions. The considered definition for the studied scales, depending on the depth of the water column and the stratification, has to be recalled as we progress in a coastal environment. In this framework, the mesoscale is defined by scales around the internal Rossby radius of deformation (~20-50 km in the mid-latitudes - Chelton et al., 1998) where the flow is adjusted under the effect of the rotation. Over the continental shelf, this internal Rossby radius of deformation decreases to values around 3-8 km (Valdivieso Da Costa M., Maze J.-P., Raynaud S., pers. comm., 2006), for example, in the Bay of Biscay. The submesoscale, as introduced by McWilliams (1985), refers to scales lower than the internal Rossby radius of deformation where the influence of the earth rotation tends to decrease in order to reach a non-rotating regime of three-dimensional turbulence (Kolmogorov, 1941). Submesoscale is then ranging from O(100)m to O(10)km over the continental shelf and in open ocean (Capet et al. 2008b, Thomas *et al.*, 2008). In the present work, we refer to (sub)mesoscale (i.e. mesoscale and submesoscale features) for processes with a length scale lower than 40km.

In the Bay of Biscay abyssal plain, coherent mesoscale structures have been identified like the long-lived anticyclonic Slope Water Oceanic eDDIES (SWODDIES) described by Pingree and Le Cann (1992) generated by slope current instabilities or quasi-stationnary eddies in the South-Eastern Bay of Biscay (Caballero et al., 2013, 2016). Following satellite altimetry based studies in the region (Caballero et al., 2008; Dussurget et al., 2011), observations of mesoscale variability have been described with higher eddy kinetic energy from December to May. However, the spatio-temporal resolution and coverage from altimetry does not allow exploring underlying processes and interannual variability at submesoscale.

In this context, after controlling the efficiency and accuracy of a coastal model with a 1km spatial resolution to reproduce the observed processes in the Bay of Biscay, the (sub)mesoscale variability at annual and interannual scales is explored as a first step to define the role of related vertical motions at small scales on long-term evolutions and associated biogeochemical production.

## 2 Numerical framework

### 2.1 Model description

Numerical simulations are based on the MARS3D model[1]. MARS3D (Duhaut et al., 2008; Lazure and Dumas, 2008) is a primitive equation model with a free surface to represent the gravity waves in the coastal area. In this finite-difference code, the primitive equations are discretized on a C-Arakawa grid centered at tracer points (Mesinger and Arakawa, 1976). The sigma coordinates are used on the vertical dimension to resolve simultaneously the shallow and deep waters. A specificity of MARS3D model is that the barotropic mode and baroclinic mode are using the same time step and the barotropic mode is resolved by an Alternating Direction Implicit method (Lazure and Dumas, 2008). Detailed equations are given in Appendix A.

---

[1] http://wwz.ifremer.fr/mars3d

[2] https://www.cines.fr/ Centre Informatique National de l'Enseignement Supérieur

[3] http://forge.ipsl.jussieu.fr/ioserver

[4] The first record is located near the bottom. Records located in the surface layer thickness (corresponding to 20 % of the

The new numerical MARS code can run without explicit viscosity (Duhaut et al., 2008). The k-ϵ turbulent closure scheme is used to model vertical mixing (Rodi, 1993).

## 2.2 Numerical experiments

The MARS3D model has already been used to investigate the Bay of Biscay and its extension to the western English Channel and focused on the validation of hydrology on the French continental shelf with a 4 km horizontal resolution and a 30 vertical levels (Lazure et al., 2009). In this new configuration, the model domain extends from the Bay of Biscay to the English Channel from 41° N to 52.5° N and 14.3° W to 4.5° E, with a 1 km spatial horizontal resolution with a time step of $\Delta t=60$ s. This configuration (called BACH1000_100lev) has 1449 x 1282 grid points and uses 100 vertical sigma levels. The vertical discretization is a generalized vertical, terrain-following, coordinate system (with $h_c=200$ m, $\theta=6$ and $b=0$; $h_c$ is the shallower depth above which we wish to have more resolution, $\theta$ and $b$ are surface and bottom control parameters, Appendix A). The bathymetry is a composite of several IFREMER Digital Terrain Models (DTM) with 100 meters resolution along the coast covering the French part of continental shelf completed by a 1 km resolution DTM covering the bay of Biscay and finally completed by a 1 nautical mile resolution from the North West Shelf Operational Oceanographic System (http://http://noos.bsh.de). Both Digital Terrain Models and mean sea level are interpolated on the grid and merged (Figure 1). Some hand editings have been performed in few key areas specially to correct spurious interpolation near the coastline. The maximum depth in the model is 5310 m. The interpolated topographies are smoothed by selectively applying a local filter to reduce the r-factor to below 0.25 ($r=\Delta h/2h$ where if the depth of the water column; Haidvogel and Beckmann, 1999). River runoffs are provided from 95 chronological records (daily measurements and climatology for past years when no observations are available) located on the Spanish, French, Netherlands, British and Irish coasts.

Initial conditions for temperature, salinity, sea surface height, baroclinic and barotropic velocities (calculated from baroclinic components) are derived from a DRAKKAR global configuration named ORCA12_L46-MJM88 (Molines et al 2014). At the open-ocean boundaries, same variables as initial conditions are used with adaptive boundary conditions in a sponge layer on North, South and West boundaries (Marchesiello et al. 2001). The sponge layer width is 20 km-width and the maximum horizontal viscosity/diffusivity values are 100 $m^2$ $s^{-1}$ and zero outside the open boundary layers. The tide, 14 harmonics constituents, is imposed along the boundaries using the FES2004 ocean tide atlas (Lyard et al., 2006).

The atmospheric forcing, which drives the simulation presented here, is provided by the ERAInterim produced by the European Centre for Medium-Range Weather Forecasts (ECMWF, Berrisford et al, 2011). Using the 2-meter air temperature, atmospherical pressure, relative humidity, rain and cloud cover from ERAInterim data, indirect calculation of the different components of the air-sea heat exchange are computed by several bulk formulae (from Lazure et al., 2009). The simulation starts from January, 1[st], 2001 and covers a 10 year period until December, 31[st], 2010. For interannual analyses, a spin-up of two years is taken into account to setup an established seasonal cycle in the circulation even in the open ocean constrained by large-scale solution forced in open boundary conditions. The analysed period is then running from 2003 to 2010. The BACH1000_100lev configuration is implemented on the Tier-1 supercomputer machine OCCIGEN provided by GENCI and hosted at CINES[2]. The supercomputer Occigen with a performance peak of 2.1 Pflops encompasses 2106 dual-socket nodes Intel@Xeon Haswell cadenced at 2.6GHz. Twelve cores are present on each socket. This numerical experiment was part of the "Big Challenges program" conducted during the VRS period from December 2014 to January 2015. Using a domain decomposition technique, the computational domain is split into 558 subdomains leading to the same number of MPI tasks with 12 OpenMP threads each. This hybrid MPI/OpenMP application

---

[2] https://www.cines.fr/ Centre Informatique National de l'Enseignement Supérieur

runs on 6696 cores and produces daily averaged outputs using the input/output server XIOS[3] specially implemented in MARS3D for this configuration.

## 3. Bay of Biscay features from spatial high resolution simulation

Before exploring (sub)mesoscale features in the Bay of Biscay, the ability of the numerical experiment to reproduce known processes in the region needs to be evaluated. Following a general view of the modelled fields, a few key diagnostics on the hydrology and the circulation are presented.

### 3.1 Sea Surface Temperature and Salinity

The numerical experiment is validated using remotely sensed Sea Surface Temperature (SST). Based on SEVIRI Sea Surface Temperature remotely sensed data (METEOSAT SST provided by OSI-SAF belong to EUMETSAT with ~2km spatial resolution), modelled fields are evaluated. In Figure 2a, the mean bias between observed and modelled SST over 2010 ( $\langle SST_{bias} \rangle_{lon,lat,t} = \left( \dfrac{1}{(N_i N_j N_t)} \displaystyle\sum_{i=1,j=1,t=1}^{N_i,N_j,N_t} SST_{i,j,t}^{\mathrm{mod}el} \right) - \left( \dfrac{1}{(N_i N_j N_t)} \displaystyle\sum_{i=1,j=1,t=1}^{N_i,N_j,N_t} SST_{i,j,t}^{observations} \right)$ where $N_i N_j N_t$ is the

number of grid points in space and time) at the end of the experiment shows an averaged underestimation of the temperature by the model (-0.24±0.28°C over the domain). These biases do not exceed 1.75 °C. Such large values are obtained in two regions. First, in the Ushant front region (around 47.5° N - 49° N and 5° W - 6° W - Le Boyer et al., 2009), the model underestimates the SST. This bias can be explained by the variability of the Ushant front, developed during stratified seasons, which remains complex to reproduce (Renaudie et al., 2011; Pasquet et al., 2012). The second main bias exceeding 1.5 °C is located along the western Spanish coast. The shape of the bias is typical of upwelling extent in this region. In this case, the coarse atmospheric forcing resolution can be emphasized as the major error source. The Figure 2b shows the temporal variation of the spatial averaged bias ( $\langle SST \rangle_{lon,lat}(t) = \dfrac{1}{(N_i N_j)} \displaystyle\sum_{i=1,j=1}^{N_i,N_j} SST_{i,j,t}$ where $N_i N_j$ is the number of grid points in space) and the shading around the curves represents the spatial standard deviation ( $\sigma_{SST}(t) = \sqrt{\dfrac{1}{(N_i-1)(N_j-1)} \displaystyle\sum_{i=1,j=1}^{N_i,N_j} (SST_{i,j} - <SST>_{lon,lat}(t))^2}$ ). At this regional scale and on average over the full domain, we do not observe significant permanent bias in the simulation. Both simulation and observation have the same average and temporal variations in surface temperature with a developed seasonal cycle with maximum temperature in August and September and the coldest waters end of winter in March. The largest differences can be noticed during the onset of the seasonal stratification in May-June.

After this first overview on SST, two contrasted dates (summer and winter) are displayed in Figure 3 for SST and Sea Surface Salinity (SSS). In summer (Figure 3a), the model reproduces the warm pool in the South-Eastern part of the Bay of Biscay with temperature exceeding 21 °C (Lazure et al., 2009). In front of the Brittany (48.2°N / 5.6°W), the position of the Ushant tidal front (Le Boyer et al., 2009; Renaudie et al., 2011; Pasquet et al., 2012) with cold waters in the vicinity of the coast and warmer water outside the front is reproduced by model simulations. In winter, colder (Figure 3b) and fresher (Figure 3d) waters above the inner shelf related to river plume extent do not exceed 9 °C and 34.8.

Furthermore, on this Figure 3, a turbulent activity (eddies, filaments) can be noticed during summer and winter in the deeper region but also over the continental shelf.

---

[3] http://forge.ipsl.jussieu.fr/ioserver

As a more focused illustration, freshwater exports in the open ocean, as described in Reverdin et al. (2013), appears in the present experiment (Figure 4). The elongated freshwater filaments extending to the Southwest in the Southern part of the Bay of Biscay (44°N / 3.5°W) are representing an observed signal of cross-slope exchanges. Reproducing these exports is a significant step forward in our simulations, thanks to the higher spatial resolution (1 km versus 4 km in previous experiments). Indeed, the spatial resolution appeared as a key issue to better resolved these exchanges between the continental shelf and the open ocean.

## 3.2 Vertical hydrological structure

The hydrological content of the simulation is evaluated through comparisons with available observations in 2010 from the CORA-IBI (COriolis ocean database for ReAnalysis - Ireland-Biscay-Iberia) database. Considered vertical profiles can be divided into two sources, ARGO (Argo, 2000; Riser et al., 2016) profiles in the open ocean and RECOPESCA (Leblond et al., 2010; Charria et al., 2014; Lamouroux et al., 2016) profiles on the continental shelf. Deeper profiles, from ARGO floats, allow analysing 2000 m depth profiles during every diving cycle. On the continental shelf, based on voluntary vessels, the RECOPESCA project provides observations of the fisheries activity (effort and catches) and the environment (temperature and salinity). Dedicated sensors are implemented on the fishing gears and allow sampling vertical hydrological profiles.

The figure 5 shows the difference between observed and modelled profiles for the year 2010 in temperature and salinity in top layers. In temperature, the model reproduces the vertical structure with a small average misfit of 0.015°C and a 0.45°C Root Mean Square Error (RMSE) between 0 and 20 m depth (Figure 5a), -0.11°C (1.23°C for RMSE) between 20 and 40 m depth (Figure 5b), and 0.25 °C (1.17°C RMSE) between 40 and 100 m depth (Figure 5c). This misfit can be observed in distribution in figure 5. Following the uncertainties around the thermocline (i.e. few meters of differences in the thermocline depth will induce large difference between model and observations) the misfit distribution is larger below 20 m depth. Similar behaviour is observed in salinity with the larger spread for the layer 20-40 m depth. In salinity, the average misfit is smaller at surface (-0.024 for the layer 0-20 m depth, RMSE=0.27, Figure 5d) and above 100 m depth (0.025 for the layer 40-100 m depth, RMSE=0.28 Figure 5e and 5f), than between 20 and 40 m depth where the average difference is larger (0.176, RMSE=0.59). Considering the RMSE, we confirm that the maximum of error is located in mid-depth layers (20-40m) and can be locally important. Part of the error can be attributed to the colocation approach assuming that we will reproduce the same features at the same time and place in the simulations, but choices for the configuration (e.g. smoothed bathymetry, coarse atmospheric forcings) can contribute to increase the observed error between model and local *in situ* observations. However, following the distributions, with biases of different sign following the depth, no systematic bias exists in the numerical experiment.

## 3.3 Bay of Biscay general circulation

Concerning the general circulation in the region, three levels of comparisons are detailed. As a synoptic view, the seasonal circulation in the surface layer is computed to be compared with existing climatologies (e.g. Charria et al., 2013). Then, to highlight circulation patterns occurring at short time scales as poleward jets over the continental shelf (e.g. Batifoulier et al., 2012; Kersalé et al., 2015) and the vertical structure of the currents, modelled fields are compared with ADCP (Acoustic Doppler Current Profiler) observations during ASPEX campaign (Le Boyer et al., 2013; Kersalé et al., 2015) and offshore Arcachon Bay during ARCADINO campaign (Batifoulier et al., 2012).

At seasonal scale, the Figure 6 shows the circulation integrated over the first 50m depth for the whole simulation. This average circulation over 10 years can be compared with the climatology (processed from observation from 1992 to 2009) derived from drifters in Charria et al. (2013). In winter (Figure 6a), the contrasted velocities with weak current over the

continental shelf and more intense structures in the open ocean clearly appear. The poleward slope current with values lower than 10 cm s$^{-1}$ is reproduced. In spring (Figure 6b), the reversal of circulation with an equatorward slope current is simulated. This circulation remains sustained in summer (Figure 6c) with a reinforcement of equatorward currents over the continental shelf. Following wind regime evolution and the transition period in September-October (SOMA seasonal response - Pingree et al., 1999), autumn circulation (Figure 6d) highlights on average the poleward slope current close to 500 m isobath. These average circulation features are then in agreement with the drifter-derived seasonal climatology (Charria et al., 2013).

Another source for validating the modelled circulation comes from ADCP deployments in the region. During the ASPEX project, 10 current-meter moorings were deployed from July 2009 to August 2011. The mooring location was distributed over the continental shelf and the upper section of the shelf break. Mooring features and observations from the project are described and analysed in Le Boyer et al. (2013) and Kersalé et al. (2015).

Two ASPEX ADCP moorings have been selected to compare evolution of current velocities with modelled fields: #4 on the continental shelf and #10 on the continental slope (see positions on Figure 1). We can notice that the length of the considered time series for comparison is both limited by the duration of the numerical experiment (2001-2010) and the technical issues in data sampling (lack of measurements for the end of ASPEX10 time series). Two dimensional linear spatial interpolation on model outputs velocity components at each sigma levels is made on the geographical position/location of the mooring. Then, the zonal and meridional components of modelled velocities are projected on along-shore and cross-shore component at each sigma level. In the aim to compare the depth-averaged velocity on both model outputs and in situ data, vertical integration of the two velocity components is made on almost the whole water column[4]. Vertical integration of the model outputs is also made on the water column between the minimum and maximum depths defined previously.

In Figure 7, the modelled and observed currents are represented. A general agreement following the current directions and amplitudes is observed with correlations between 0.6 and 0.69 for along-shore components. For the cross-shore component, representing the less intense currents, in ASPEX4 (Figure 4b), the direction of the current is well reproduced but the amplitude remains generally smaller in simulations (RMSE=0.024 m s$^{-1}$). At ASPEX10 (Figure 4d), the cross-shore weak circulation is not reproduced, with a correlation between model and observation equal to 0.09, due to the mesoscale circulation in this area (e.g. Solabarrieta et al., 2014; Caballero et al., 2016). The agreement between observations and numerical simulation is improved for dominant along-shore currents. Indeed, amplitudes are very similar in both ASPEX sites (except during autumn 2009, Figure 4a and 4c). The direction and direction changes are also very well reproduced (the correlation for ASPEX4 is equal to 0.6 and for ASPEX10 to 0.69), even at high frequency, which was not expected following the coarse atmospheric forcings used for the simulation.

Other comparisons have been performed with an ADCP mooring during the ARCADINO experiment. This mooring located on the Aquitaine shelf (South of 45°N, Figure 1) has been used to highlight poleward coastal jets up to 32 cm s$^{-1}$ (Batifoulier et al., 2012). Similar events are modelled in our numerical experiments with smaller amplitudes (Figure 8). In 2008, a poleward along-shore current appears around the 15th August 2008 (Figure 8) in observations. From in situ ADCP measurements (Batifoulier et al., 2012), it also appears a velocity maximum between the 16th and 20th August 2008. In the modelled fields, the jet is reproduced but velocities are weaker and the event starts earlier in the simulation. The jet is also deeper in the model (20-40 m depth with maximum velocities ~16 cm s$^{-1}$) than in observations with a maximum above 30 m depth. When model forcings are explored, we explain this event with similar conditions to those observed in Batifoulier et al.

---

[4] The first record is located near the bottom. Records located in the surface layer thickness (corresponding to 20 % of the mooring depth) have been removed due to noisy measurements.

2012. Indeed, westerly winds are blowing from the 6th to the 8th August (with intensities 8 to 12 m s$^{-1}$) along the Spanish coast to setup the circulation resulting in the poleward jet following the explained process in Batifoulier et al. (2012).

These illustrations of the modelled fields and comparisons with available observations show the ability and the limits of our numerical experiment to reproduce the coastal ocean dynamics at high resolution in the Bay of Biscay. Based on these fields

the interannual variability at (sub)mesoscales can be explored.

## 4. Interannual variability of (sub)mesoscale instabilities in surface layers

The present study aims to characterize the interannual variability of the (sub)mesoscale dynamics and discussing the possible processes explaining this variability. Before considering these interannual scales, the seasonal features are described for a

given year.

### 4.1 Seasonal scale

To explore the (sub)mesoscale activity, the vertical component of the relative vorticity (referred as relative vorticity) has been first analysed. From these dynamical fields, we can infer the intensity of rotating structures and their spatial distribution.

In figure 9, the surface relative vorticity from analysed simulations at different contrasted time steps is represented. From these maps, different patterns can be noticed. First, the contrast between the deep open ocean and the shallow continental shelf is clearly visible for the different periods. In summer 2003 (Figure 9a), over the continental shelf, the internal waves are observed in the northern part of the domain spreading from the shelf break (around 47.5°N-48°N / 7°W-5°W). In the southern part of the continental shelf  (South of 48°N), small structures related to local drivers (*e.g.* edge of region of

freshwater influence, wind bursts - Yelekci, O. et al., pers. comm., 2016) are developed. These structures can be seen through large relative vorticity values over the outer part of the continental shelf between the 100m isobath (Figure 6) and the shelf break (Figure 1). At the opposite, in winter (Figure 9b), small-scale features are more concentrated in the inner shelf (the first half of the continental shelf closer to the coast with water shallower than 100m depth) under the influence of large winter river inputs (*e.g.* mainly from Loire and Gironde rivers).

When we consider the open ocean over the abyssal plain, contrasted situations with structures with smaller relative vorticity in summer (Figure 9a) and more intense small vortices in winter (Figure 9b) are clearly observed. Smaller features (eddies and filaments with spatial scales lower than 40km) are fully developed in winter. In summer, typical spatial scales are larger in than in winter. More large scales vortices are simulated during this season. The spatial spectral analysis over the domain (Figure 11) confirms the largest small-scale (< 50 km wavelength) variance peaks in winter (maximum in March) and the

minimum variance at small-scale in summer (July).

Based on Figure 10 representing the years 2004 and 2005, a picture of the annual evolution of the relative vorticity intensity can be drawn considering a spatial average of the absolute relative vorticity over the region highlighted in Figure 9 (yellow rectangle). Based on the spatial average integrated over 150 m depth[5] (Figure 10a), a maximum is observed during the end of

winter (February - March) followed by a period (June to September) corresponding to a minimum of averaged relative vorticity. The horizontal patterns (Figures 10b and 10c) associated with these average time series confirm the larger range of relative vorticity values related to small scales structures. In summer (Figure 10c), intensity of eddies is decreased with larger scale features (*e.g.* structures with a length scale larger than 50 km). This decrease in the intensity of the modelled

---

[5] This depth (150m) has been defined to include most of mixed layer depth in winter. As it is used for the whole time series (including stratified seasons), the maximum mixed layer depth (around 200 to 250m on average) has not been taken as a reference.

field can also be described through the surface relative vorticity spectra computed from each month (Figure 11). These spectra have been computed every day over the limited domain (yellow rectangle Figure 9) using a 2D Fast Fourier Transform and then averaged over the considered month. These spectra clearly show the seasonal variation of the variance with a maximum in March and a minimum in July. An increase in the variance of small scales (lower than 50 km) is also observed through a change in the curve slope observed in November, January and March compared with May, July and September months.

Following the relative vorticity fields (i.e. related to vortices, fronts, filaments), vertical motions can also be explored. The role of structures at (sub)mesoscale on vertical mixing can be highlighted by the exploration of vertical velocities (Significant vertical velocity patterns are mainly at submeso- and meso-scales). Indeed, in Figure 12a, a similar seasonal cycle with relative vorticity is observed with a maximum of integrated vertical velocity end of winter (February - March) and a minimum in summer (June to September). Based on the spatial patterns of the vertical velocity fields (Figures 12b and 12c), intense and small structures are observed end of winter (Figure 12b) developed with small typical length scales. In summer, positive and negative vertical velocity patterns are more elongated related with aggregated patterns (Figure 12c) and less activity at small-scale.

A vertical signature of the fluctuations in the (sub)mesoscale regimes can be inferred from the (sub)mesoscale component of the vertical buoyancy flux ($w'b'$ where $w$ is the vertical velocity and $b$ the buoyancy) computed following:

$$\begin{cases} w = \overline{w} + w' \\ b = \overline{b} + b' \end{cases}$$

$$\text{with } b = -g\frac{(\rho - \rho_0)}{\rho_0}$$

where $\overline{w}$ and $\overline{b}$ are filtered field using a 2D convolution with a Hanning window of 40 km length scale. $w'b'$ is then representing spatial scales smaller than 40 km.

The diagnostic ($w'b'$) translates the conversion rate of available potential energy to eddy kinetic energy (e.g. Boccaletti et al. 2007; Fox-Kemper et al. 2008), which tends to be maximal in the Mixed Layer in the case of vertical velocities related to Mixed Layer Instabilities (Boccaletti et al. 2007; Stone 1966, 1970). In Figure 13, the vertical profile of $w'b'$ averaged over the studied subdomain during winter season (January to March) shows a maximum (reaching, on average, 3.1 $10^{-10}$ m$^2$ s$^{-3}$) in surface layers corresponding to the mixed layer.

Following the seasonal description, 10 years of high-resolution simulations allows considering the interannual variations.

**4.1 Interannual scale**

The different regimes modelled in 2004 and 2005 are also observed during the whole simulated period (2003-2010 - the first two years are not taken into account considering a spin-up period). Indeed, in Figure 14a (vertical velocity) and 14b (relative vorticity), a maximum appears generally end of winter at the same time for both quantities. The intensity of the maximum displays interannual fluctuations with larger values in 2004 (only for vertical velocity), 2005, 2006, 2009 and 2010. At the opposite, 2003, 2007 and 2008 are characterized by a weaker (sub)mesoscale activity. The maxima are in phase with the coldest period in temperature and associated with the larger anomalies compared with the averaged annual cycle (Figure 14c) before the spring warming and the beginning of seasonal stratification. The most extreme vertical velocities are simulated during the winter 2005 with a peak begin of March 2005. At the opposite positive anomalies in temperature are modelled from September 2007 until May 2008. For both winter before (begin of 2007) and during this period (winter 2008),

minimum vertical velocities and relative vorticity is observed over the 8 years period. In 2009, the winter situation is coming back to cold sea temperature anomalies related with more intense vertical velocities and relative vorticity.

As we consider an area not under direct influence of major river runoffs (far from the slope dynamic barrier), the salinity (Figure 14d) does not exhibit a regular seasonal cycle. Indeed, main sources of freshwater in the Bay of Biscay are coming for river discharges. These discharges follow a seasonal cycle with a maximum flow end of winter not simulated over the analysed domain. Furthermore, the evaporation-precipitation budget (related to the more intense and frequent depression in winter) does not induce large variations at seasonal scales in the region but fluctuates interannually depending the atmospheric conditions.

The role of the different spatial scales in this interannual variability is explored through the analysis of the slope of the power spectrum of surface relative vorticity. The Figure 15 shows the maximum slopes (larger than $k^{-0.4}$) occurring in winter (from November - year-1 to March year). At the opposite, slopes values are very small (between $k^{-1.2}$ and $k^{-1.4}$) in spring with a minimum in May or June. The interannual variability of this minimum is limited and values are very similar following the year. Concerning the winter maximum, the value is decreasing with time but the limited number of simulated years does not allow concluding to the significance of this trend. The monthly seasonal cycle is very stable every year. However, we can notice that in 2004, high slope values are reached earlier (in November) than during the other years (December, January or February). The interannual variability of the spectral slope gives then an overview of the evolution of spatial scales distribution.

## 5. Discussion

Model simulation, validated with available observations, exhibits a seasonal cycle related to small-scale features in the deep part of the Bay of Biscay. This region, despite low level of eddy kinetic energy (e.g. Caballero et al., 2008; Charria et al., 2013), is the location of development of mixed layer instability dynamics similar to those observed in western Pacific Ocean (e.g. Sasaki et al., 2014), the western North Atlantic (e.g. Mensa et al., 2013; Callies et al., 2015), and the eastern North Atlantic (e.g. Thompson et al., 2016). Following the analogy, the features from mixed layer instabilities (Boccaletti et al. 2007) are confirmed by the maximum of activity simulated at the end of winter when vertical buoyancy fluxes at (sub)mesoscale are the most intense and with a maximum of conversion rate between available potential energy and eddy kinetic energy at (sub)mesoscale in the mixed layer depth[6] (Figure 13). These instabilities drive to a conversion in kinetic energy of the stored potential energy in winter and lead to reinforce the seasonal stratification.

Therefore, in a realistic modelling framework, these results corroborate the suitable spatial (1 km) and vertical resolutions (100 sigma levels) to solve the (sub)mesoscale realistic features resulting from mixed layer instabilities. Indeed, Soufflet et al. (2016), based on ROMS simulations in a baroclinic jet test case, showed the sensitivity of the vertical buoyancy flux to the spatial resolution (20 km, 10 km, 5 km, and 2 km) with a maximum mixed layer buoyancy flux for the higher resolution model. In the present study, the reproducibility of the results balancing between winter unstable field and summer smoothed mesoscale activity after 10 years of simulation further shows the interest of the O(1 km) scale in regional modelling. Previous interannual experiments with 4km spatial resolution (not shown) also confirm the improvements.

The system described in the Bay of Biscay is then following a scheme where end of winter mixed layer instabilities will feed the eddy kinetic energy in the region. However, interannual fluctuations are clearly visible (Figure 14) and can have an effect on the intensity of instabilities. A first link has been established between the winter mixed layer depth and the submesoscale activity. Indeed, Figure 16a, representing the averaged mixed layer depth in the studied region, is correlated with the evolution of the relative vorticity and associated vertical velocities (Figures 14a and 14b). The maximum intensity of vertical

---

[6] The criterion selected for the mixed layer depth is a threshold value of density from a near-surface value at 10 m depth equal to 0.03 Kg m$^{-3}$ following de Boyer Montégut et al. (2004).

velocities is related to the maximum depth of the mixed layer. This relationship can be explained by the amount of available potential energy stored following these deep mixed layers. Following the potential impact of such fluctuations (maximum average vertical velocities can be doubled following the considered year) on the mixing and then on systems under this pressure (e.g. biologeochemistry), identifying the source of such variability becomes a key point to forecast seasonal small scale dynamics. A first driver potentially explaining deeper mixed layer depth some years is the mechanical energy input (e.g. Duhaut and Straub, 2006; Huang et al., 2006; Elipot and Gille, 2009) related with the wind stress and the surface ocean velocity (the surface ocean velocity effect is generally smaller than the wind stress impact). Variations of this large scale source of energy has been explored in the Bay of Biscay and does not explain the interannual variations of the mixed layer depth in the region (not shown).

The alternative source of convective processes deepening the mixed layer depth in winter is the heat fluxes (mostly latent and sensible heat fluxes in winter in the region following Somavilla et al., 2011). During the simulated period, the extremely cold and dry winter in 2005 (Somavilla et al., 2009, 2011, 2016) explains the deepest average mixed layer depth over the domain. This winter was very specific with dominant Northerly wind (Figure 17) advecting cold air in the Bay of Biscay. This cold air mass influences the air-sea temperature gradient and then the associated heat fluxes. This extreme winter is associated to the largest vertical buoyancy flux at (sub)mesoscale (Figure 16b). Following the same behaviour, the years 2009 and 2010 reaches also deep mixed layer depth maximum (deeper than 250m - Figure 16a) associated with an intense associated vertical buoyancy flux. Similarly, the year 2010 is associated with an important occurrence of Northerly winds (Figure 17). The modelled deep mixed layer for these years were observed by Hartman et al. (2014) from in situ Argo vertical profiles in the Bay of Biscay. These specific years (2009 and 2010) were also associated to cold winters.

At the opposite, 2007 and 2008 shallower mixed layers (Figure 16a) associated with an eroded maximum of vertical buoyancy flux at (sub)mesoscale (Figure 16b) are related to warm winters causing warming of the surface ocean and a decrease in winter mixing. Indeed, during the winter 2007, the surface air temperature was probably the highest record during the past 500 years (Luterbacher et al., 2007).

The winter 2006 is an intermediate state due to remaining effect of sea surface temperature anomaly during winter 2005 (Dummousseaud et al., 2010).

The analysis can be extended to the distribution of the dominant spatial scales. Based on power spectra and the evolution of the spectral slopes (Figure 15), the analysis does not show interannual evolution in the distribution of spatial scales except end of 2004 where we can observe that the maximum slope is reached in November, earlier than during other analysed years. During the whole period, slopes remains located between $k^{-0.4}$ and $k^{-1.4}$. This range is in agreement with modelling studies based on similar resolutions. For example, in Brannigan et al. (2015), spectral slopes for surface velocities for simulation with similar resolution (1km and 2km) are located between $k^{-2}$ and $k^{-4}$. Taken into account the velocity derivative in the relative vorticity, slopes from the present study are equivalent to slopes between $k^{-2.4}$ and $k^{-3.4}$ in surface velocities.

Based on these distributions, the potential impact of large-scale interannual variability on the small-scale features is mainly observed for extreme conditions (e.g. autumn2004/winter 2005) where the early decrease of the slope translates an anticipated increase of the variance at small-scales.

## 6. Conclusions

With the rise of numerical capabilities, coastal dynamics can be explored at regional scale over pluri-annual periods keeping a high spatial resolution needed to solve (sub)mesoscale. Based on a 1km spatial resolution numerical experiment over 10 years, we explored the (sub)mesoscale dynamics in the Bay of Biscay and its interannual evolutions.

Before exploring interannual variability for few kilometre scales, the ability of the model to reproduce multi-scale processes (from intermittent events to average circulation) has been shown including sustaining a coherent circulation after 10 years of simulation.

Based on these products and despite low levels of eddy kinetic energy linked with an eastern boundary circulation system, the seasonal cycle in the turbulent regimes with smaller scale end of winter and a maximum in relative vorticity and vertical velocities end of winter (in March) is shown. The source of these small-scale structures is associated to mixed layer instabilities.

Then, the investigations focused on interannual variability in (sub)mesoscale are linking the evolutions in the maximum of small-scale vertical velocities with the maximum depth of the mixed layer depth reached during the on-going winter. Differences between intensities of (sub)mesoscale activity can then be related to the winter conditions explaining mixed layer dynamics. Cold winters are characterized by deeper mixed layer depth (2005, 2009 and 2010) with the coldest winter in 2005, which induced a shift in the North Atlantic heat budget and circulation (Somavilla et al., 2016). These cold winters are associated with more intense baroclinic instabilities inducing vertical velocities at (sub)mesoscale and an early increase of small-scale variance (November in 2004). At the opposite, years 2006 to 2008 represent warm winter (with the warmest in 2007), shallow mixed layer and a weak generation rate of eddy kinetic energy.

Therefore, this experiment shows a straight impact of large-scale ocean-atmosphere heat fluxes on the intensity of (sub)mesoscale activity in a region under coastal influence. This new insight in understanding (sub)mesoscale in coastal regions, thanks to high-resolution numerical modelling, will contribute understanding small-scale fluctuations in biogeochemical production.

**Acknowledgements**

This study is part of the LEFE/GMMC project ENIGME. Model experiments have been performed with one of GENCI ( french Grand Equipment Company National Supercomputing) computational resources administrated at CINES (National Computing Center for Higher Education). It costs 3.6 millions of cores-h. We would like to thank Arnaud Le Boyer and Pascal Lazure for providing ASPEX data. Remotely sensed Sea Surface Temperature data are provided by OSI-SAF (http://www.osi-saf.org/) belongs to EUMETSAT. Data processing has been performed using open-source Python library VACUMM. We also thank Bernard Le Cann, Louis Marié and Christophe Maes for the insightful discussions. Special thanks are due to Dr. Brannigan and an anonymous reviewer for very constructive comments.

**Appendix A**

In the MARS3D model, the set of primitive equations (Lazure and Dumas, 2008) is obtained based on usual assumptions (Boussinesq and shallow-water assumptions) in an hydrostatic framework. As the model is based on vertical sigma coordinates, equations are re-written in a sigma coordinate framework, where (Song and Haidvogel, 1994):

$$z = \zeta(1+\sigma) + h_c\sigma + (H - h_c)C(\sigma) \qquad (1)$$

With $C(\sigma) = (1-b)\dfrac{\sinh(\theta\sigma)}{2\sinh\theta} + b\left[\dfrac{\tanh[\theta(\sigma+1/2)]}{2\tanh(\theta/2)} - \dfrac{1}{2}\right]$

Where $\sigma$ is the vertical coordinate, D is the height of water column, with D=H+$\zeta$. H is the depth of the fluid at rest, $\zeta$ is the sea surface elevation. z and $\sigma$ increase upwards. The result is that at the sea surface $(z=\zeta)$ and $\sigma$=0. At the opposite, at the sea floor $(z=-H)$ and $\sigma = -1$.

We have noted the L operator as,

$$L(A) = u\frac{\partial A}{\partial x} + v\frac{\partial A}{\partial y} + w^*\frac{\partial A}{\partial \sigma} \tag{2}$$

$u$ is the zonal velocity, $v$ the meridional velocity, and $w^*$ is the vertical velocity in the sigma coordinate framework $(x,y,\sigma)$ with,

$$w^* = \frac{1}{D}(w - \sigma\frac{\partial \xi}{\partial t} - u(\sigma\frac{\partial \xi}{\partial x} + (\sigma - 1)\frac{\partial H}{\partial x}) - v(\sigma\frac{\partial \xi}{\partial y} + (\sigma - 1)\frac{\partial H}{\partial y})) \tag{3}$$

The set of primitive equations is then in Cartesian coordinates :

$$\frac{1}{D}\frac{\partial p}{\partial \sigma} = -\rho g \tag{4}$$

$$\frac{\partial u}{\partial t} + L(u) - fv = -g\frac{\partial \xi}{\partial x} - \frac{1}{\rho_0}\frac{\partial Pa}{\partial x} + \pi_x + \frac{1}{D}\frac{\partial(\frac{nz}{D}\frac{\partial u}{\partial \sigma})}{\partial \sigma} + F_x \tag{5}$$

$$\frac{\partial v}{\partial t} + L(v) + fu = -g\frac{\partial \xi}{\partial y} - \frac{1}{\rho_0}\frac{\partial Pa}{\partial y} + \pi_y + \frac{1}{D}\frac{\partial(\frac{nz}{D}\frac{\partial v}{\partial \sigma})}{\partial \sigma} + F_y \tag{6}$$

$$\frac{\partial \xi}{\partial t} + \frac{\partial Du}{\partial x} + \frac{\partial Dv}{\partial y} + \frac{\partial Dw^*}{\partial \sigma} = 0 \tag{7}$$

$$\frac{\partial DT}{\partial t} + \frac{\partial D(uT - k_x\frac{\partial T}{\partial x})}{\partial x} + \frac{\partial D(vT - k_y\frac{\partial T}{\partial y})}{\partial y} + \frac{\partial D(w^*T - \frac{k_z}{D^2}\frac{\partial T}{\partial \sigma})}{\partial \sigma} = \frac{1}{\rho_0 C_P}\frac{\partial I}{\partial \sigma} \tag{8}$$

$$\frac{\partial DS}{\partial t} + \frac{\partial D(uS - k_x\frac{\partial S}{\partial x})}{\partial x} + \frac{\partial D(vS - k_y\frac{\partial S}{\partial y})}{\partial y} + \frac{\partial D(w^*S - \frac{k_z}{D^2}\frac{\partial S}{\partial \sigma})}{\partial \sigma} = 0 \tag{9}$$

The equation of state relates density to salinity, temperature and pressure:

$$\rho = F(S,T,p) \tag{10}$$

With F is a non-linear function (not stated explicitly here, , Jackett and McDougall, 1995).

From the equation (1) and introducing the buoyancy $b = -g(\rho - \rho_0)/\rho_0$ within a sigma coordinate framework, the zonal and meridian components of the baroclinic pressure gradient ($\pi_x, \pi_y$) are:

$$\pi_x = \frac{\partial}{\partial x}[D\int_\sigma^1 b\,d\sigma] + b(\sigma\frac{\partial D}{\partial x} - \frac{\partial H}{\partial x}) \tag{11}$$

$$\pi_y = \frac{\partial}{\partial y}[D\int_\sigma^1 b\, d\sigma] + b(\sigma\frac{\partial D}{\partial y} - \frac{\partial H}{\partial y}) \tag{12}$$

Horizontal friction terms are,

$$F_x = \frac{1}{D}\frac{\partial}{\partial x}[Dv_x\frac{\partial u}{\partial x}] + \frac{1}{D}\frac{\partial}{\partial x}[v_x(\frac{\partial H}{\partial x} - \sigma\frac{\partial D}{\partial x})\frac{\partial u}{\partial\sigma}] +$$
$$\frac{1}{D}\frac{\partial}{\partial\sigma}[v_x(\frac{\partial H}{\partial x} - \sigma\frac{\partial D}{\partial x})\frac{\partial u}{\partial x}] + \frac{1}{D}\frac{\partial}{\partial\sigma}[\frac{v_x}{D}(\frac{\partial H}{\partial x} - \sigma\frac{\partial D}{\partial x})^2\frac{\partial u}{\partial\sigma}] \tag{13}$$

$$F_y = \frac{1}{D}\frac{\partial}{\partial y}[Dv_y\frac{\partial v}{\partial y}] + \frac{1}{D}\frac{\partial}{\partial y}[v_y(\frac{\partial H}{\partial y} - \sigma\frac{\partial D}{\partial y})\frac{\partial v}{\partial\sigma}] +$$
$$\frac{1}{D}\frac{\partial}{\partial\sigma}[v_y(\frac{\partial H}{\partial y} - \sigma\frac{\partial D}{\partial y})\frac{\partial v}{\partial y}] + \frac{1}{D}\frac{\partial}{\partial\sigma}[\frac{v_y}{D}(\frac{\partial H}{\partial y} - \sigma\frac{\partial D}{\partial y})^2\frac{\partial v}{\partial\sigma}] \tag{14}$$

where:

$x, y, \sigma$ Cartesian coordinates of the framework u,v et w[*] respectively zonal meridian and vertical velocity components, $H(x,y)$ absolute value of bottom position, $S, T, p$ are respectively salinity, temperature and pressure.

$f = 2\Omega sin\phi$ Coriolis parameter, $\Omega = 2\pi/86164$ rad/s earth's rotation frequency, $g$ gravity, $b = -g(\rho - \rho_0)/\rho_0$ buoyancy, $\rho = \rho(S,T,p)$ seawater density, $\rho_0$ reference density, $C_p$ sea water heat capacity, $I$ shortwave heat fluxes, $nz$ vertical eddy viscosity, $kz$ vertical eddy diffusivity, $v_x$ et $v_y$ horizontal eddy viscosity, $k_x$ and $k_y$ horizontal eddy diffusivity.

The boundary conditions are expressed as:

| Boundary conditions at the surface σ = 0 | Boundary conditions at the bottom σ =−1 |
|---|---|
| $\frac{nz}{D}\frac{\partial u}{\partial\sigma} = \frac{\tau_{sx}}{\rho_0}$ | $\frac{nz}{D}\frac{\partial u}{\partial\sigma} = \frac{\tau_{bx}}{\rho_0}$ |
| $\frac{nz}{D}\frac{\partial v}{\partial\sigma} = \frac{\tau_{sy}}{\rho_0}$ | $\frac{nz}{D}\frac{\partial v}{\partial\sigma} = \frac{\tau_{by}}{\rho_0}$ |
| $\frac{kz}{D}\frac{\partial T}{\partial\sigma} = \frac{Q_T}{\rho_0 C_p}$ | $kz\frac{\partial T}{\partial\sigma} = 0$ |
| $kz\frac{\partial S}{\partial\sigma} = 0$ | $kz\frac{\partial S}{\partial\sigma} = 0$ |
| w[*] = 0 | w[*] = 0 |

where $Q_T$ is the heat flux at the air-sea interface, $(\tau_{sx}, \tau_{sy}) = \rho_a Cd_s\|\vec{W}\|(W_x, W_y)$ are the surface stress components with

$\rho_a = 1.25 kg/m^3$
$Cd_S = 0.016$

$(W_x, W_y)$ is the wind velocity vector at 10 m above the sea surface.

$(\tau_{bx}, \tau_{by}) = \rho_0 Cd_B \|\vec{u}\| (u,v)$ are the bottom stress components with

$$Cd_B = \left( \frac{\kappa}{\ln\left( \dfrac{z + H + z_0}{z_0} \right)} \right)^2$$

where $\kappa$ refers to the Von Karman constant and $z_0$ the bed roughness.

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

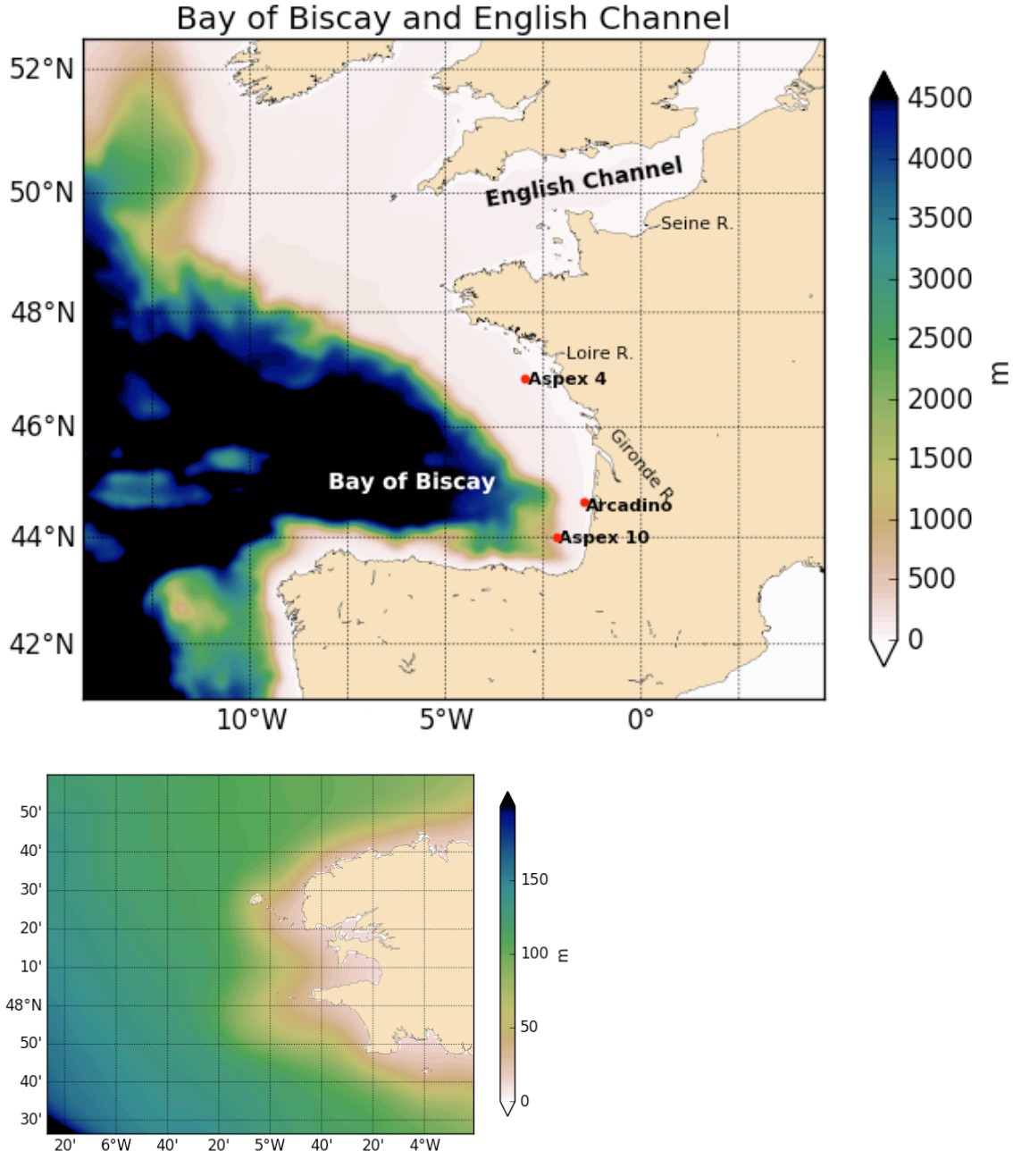

**Figure 1: Bathymetry of the modelled region (top). Red points are corresponding to the mooring sites used for model validation. A zoomed-in area around 48°N is represented in bottom figure.**

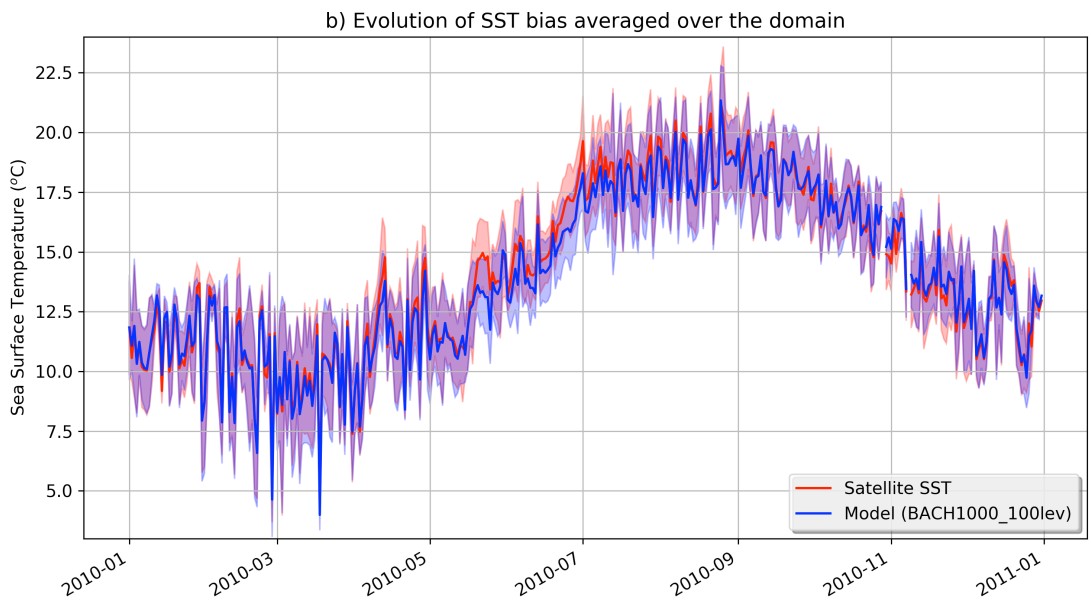

a) SST bias (model - observations) over the domain

b) Evolution of SST bias averaged over the domain

**Figure 2: Comparison between observed (SEVIRI satellite SST) and modelled (BACH1000_100lev simulation) Sea Surface Temperature (SST). a) Mean bias between model and observations for the year 2010. b) Temporal evolution of the spatial mean SST bias during 2010. The shading around the curves represents the spatial standard deviation (*i.e.* the standard deviation over the domain computed for each time step).**

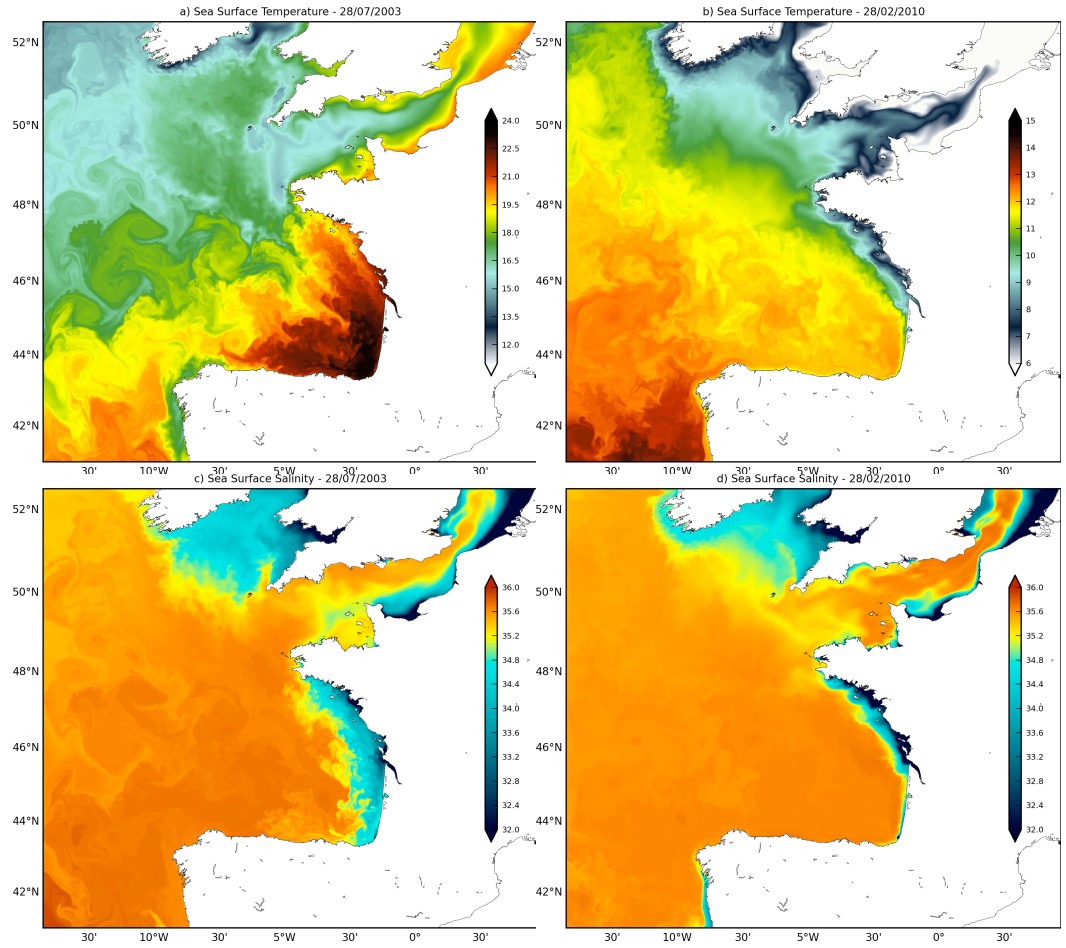

5    **Figure 3: Example of modelled (BACH1000_100lev configuration) Sea Surface Temperature (a,b) and Salinity (c,d) in summer (a,c - 28 July 2003) and in winter (b,d - 27 February 2010).**

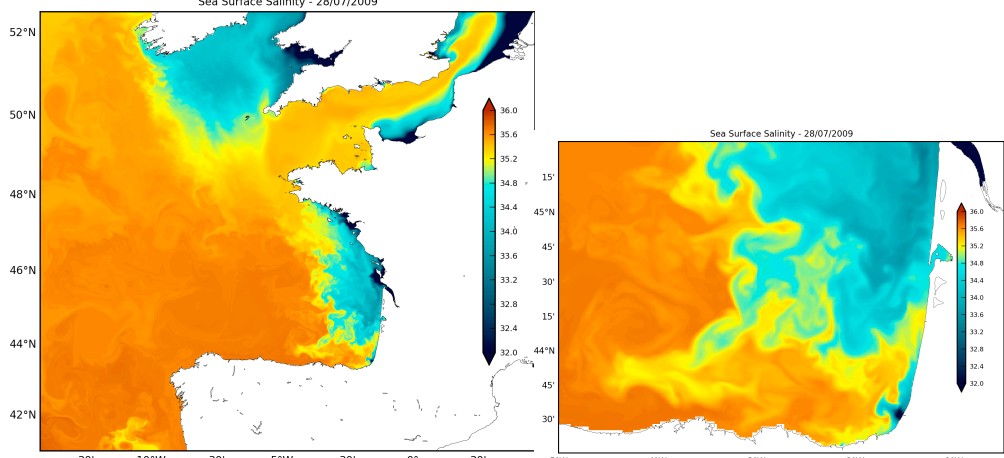

**Figure 4: Sea surface salinity (28th July 2009) during an event of freshwater export in the open ocean as described by Reverdin *et al.* (2013). Left figure is representing the full model domain and right figure is focused on the South-Eastern part of the Bay of Biscay to highlight the freshwater export around 44°N and 3.5°W.**

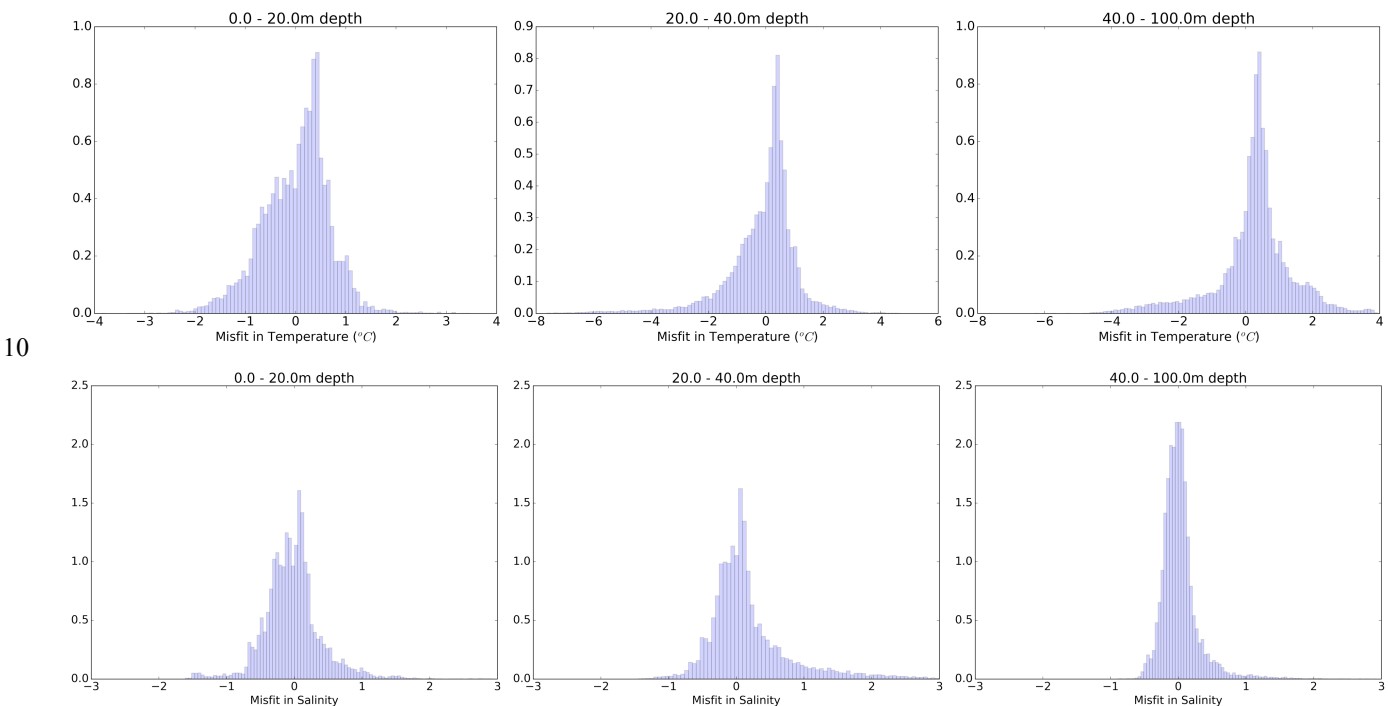

**Figure 5: Normalized Distribution of the misfit (modelled - observed) in Temperature (top) and Salinity (bottom) from RECOPESCA and ARGO in situ profiles (only for profiles deeper than 100m-depth) for three vertical layers: 0-20m depth (left), 20-40m depth (middle), and 40-100m depth (right). The integral of the histogram sum to 1.**

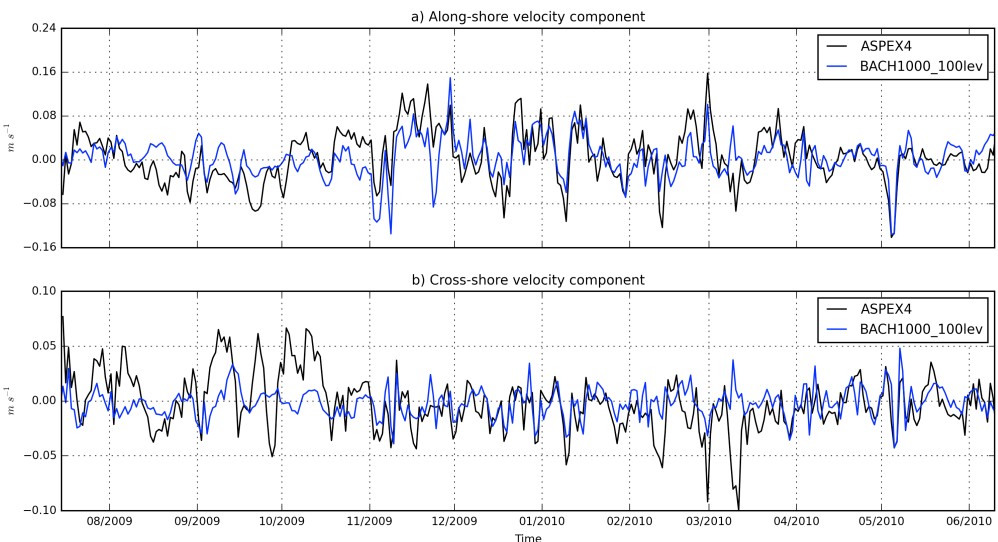

**Figure 6: Average modelled seasonal circulation (a) Winter, b) Spring, c) Summer, d) Autumn) for surface layers (0-50m depth) over the period 2001-2010 (for clarity purpose, 1 over 50 grid points are plotted). Gray lines are representing 500 m, 200 m, 100 m and 50 m isobaths.**

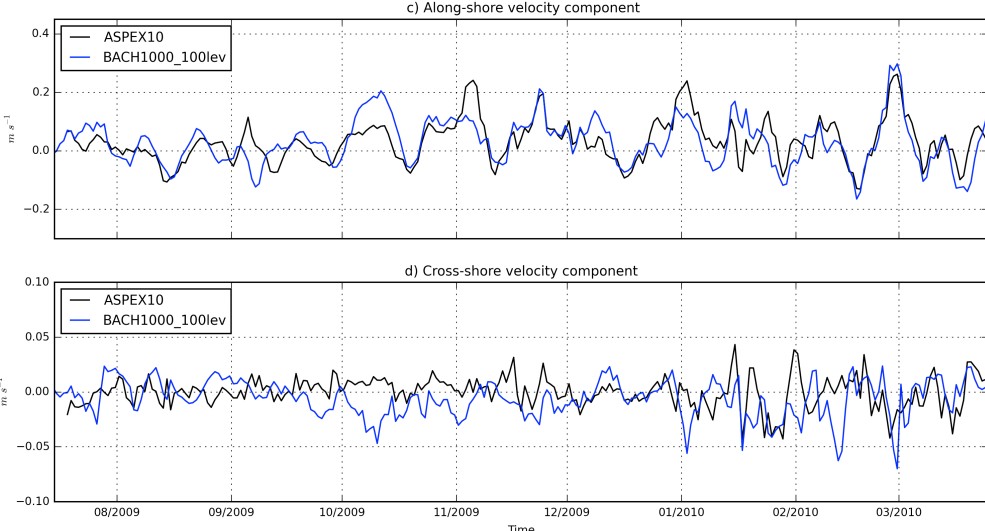

**Figure 7: Comparison of the 1-day mean and depth averaged along-shore and cross-shore velocity component between ADCP measurements (black) and BACH1000_100lev currents (blue) at the location of  ASPEX4 (fig. 7a and fig. 7b) above the continental shelf and ASPEX 10 (fig. 7c and fig. 7d) above the continental slope. The orientation of the along-shore and cross-shore component is relative to the bathymetry.**

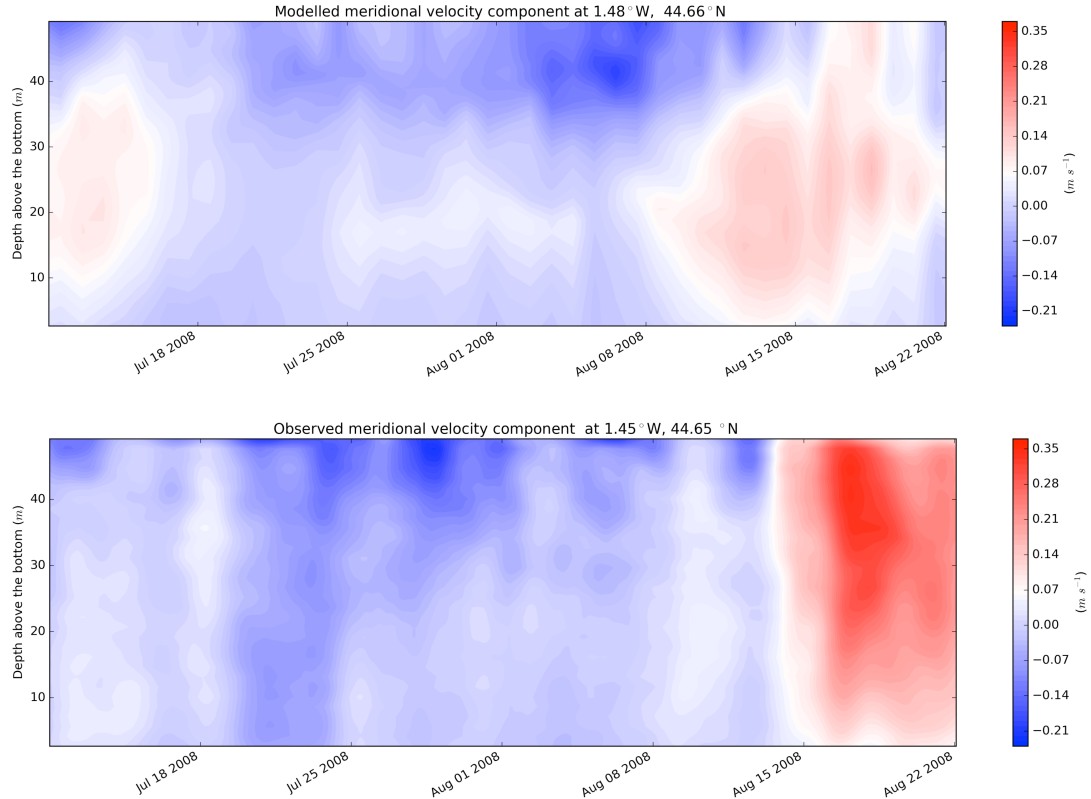

**Figure 8: Evolution of meridional velocity component in m s⁻¹ of the BACH1000_100lev model during July-August 2008 (top) at ARCADINO ADCP location. ADCP observations for the same periods are represented (bottom).**

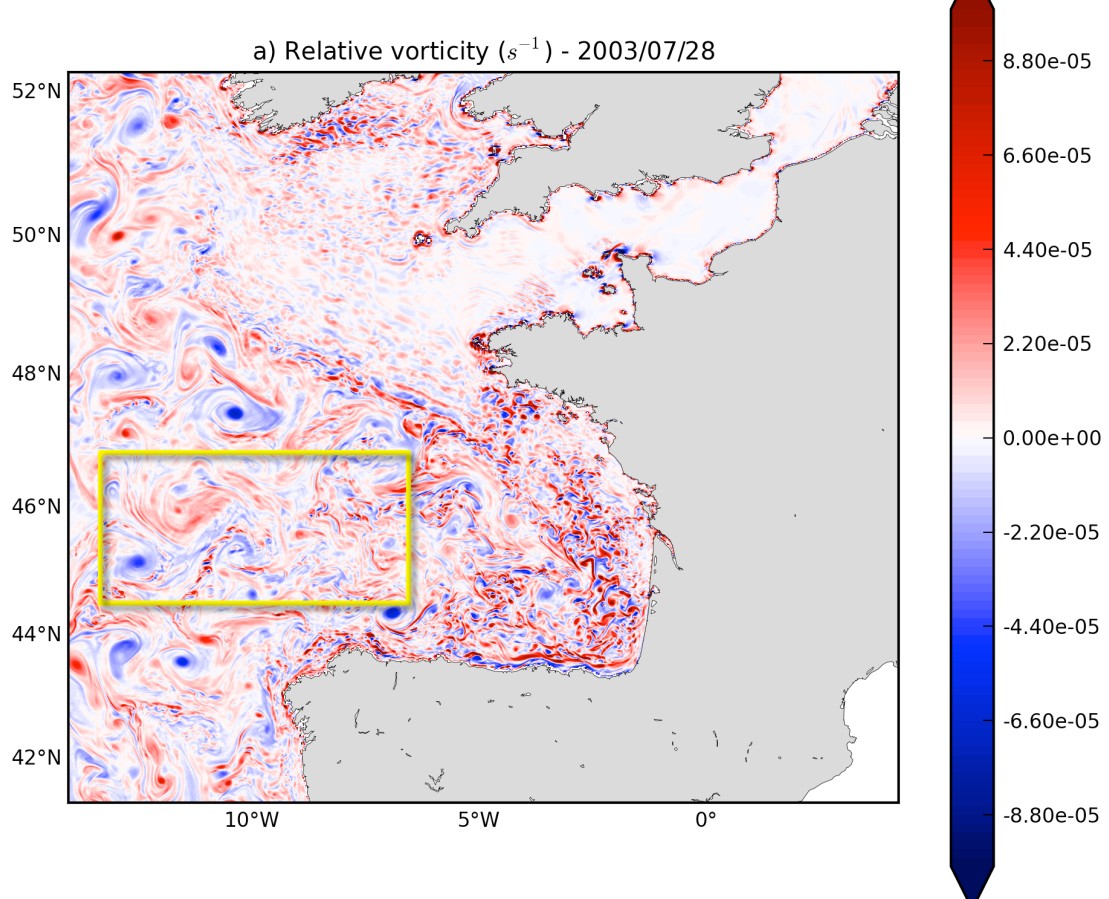

a) Relative vorticity ($s^{-1}$) - 2003/07/28

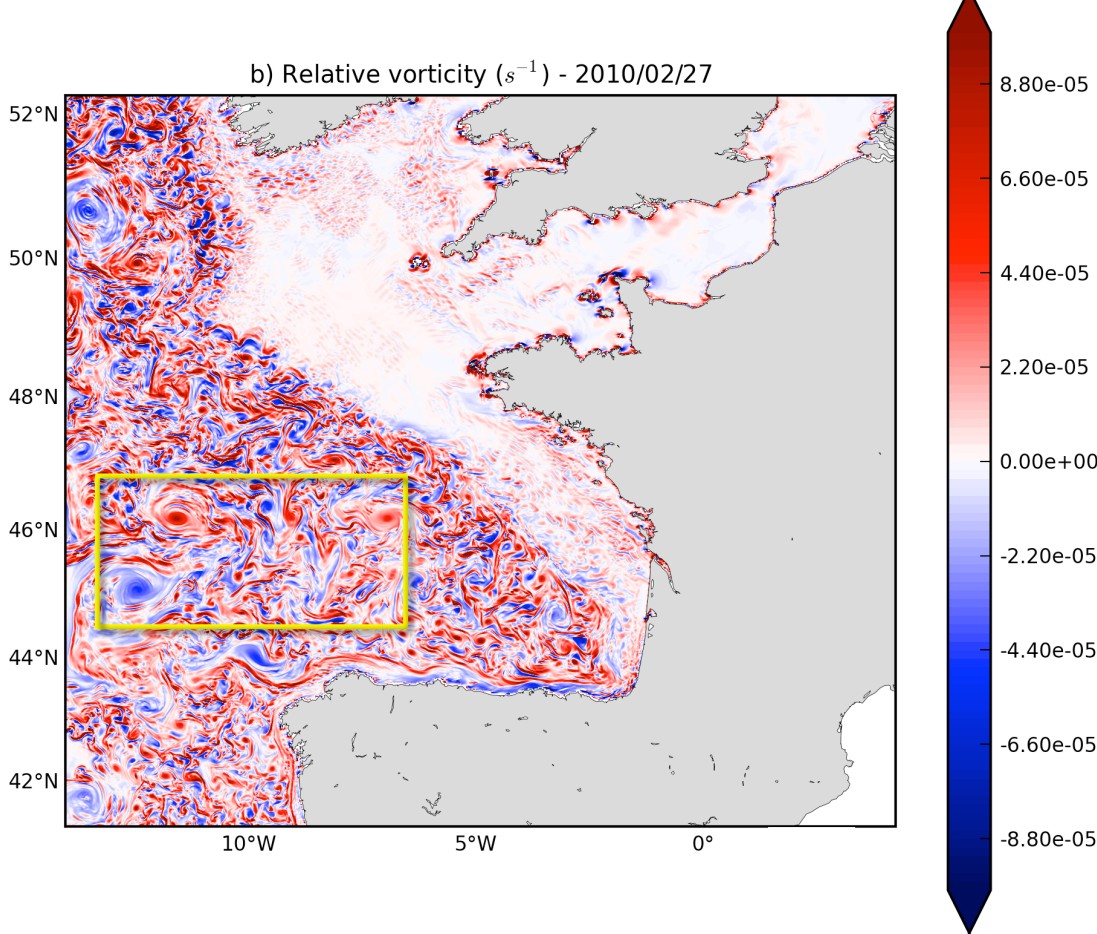

**Figure 9: Surface modelled relative vorticity for the 28th July 2003 (a) and the 27th February 2010 (b). The yellow rectangle limits the targeted region for diagnostics.**

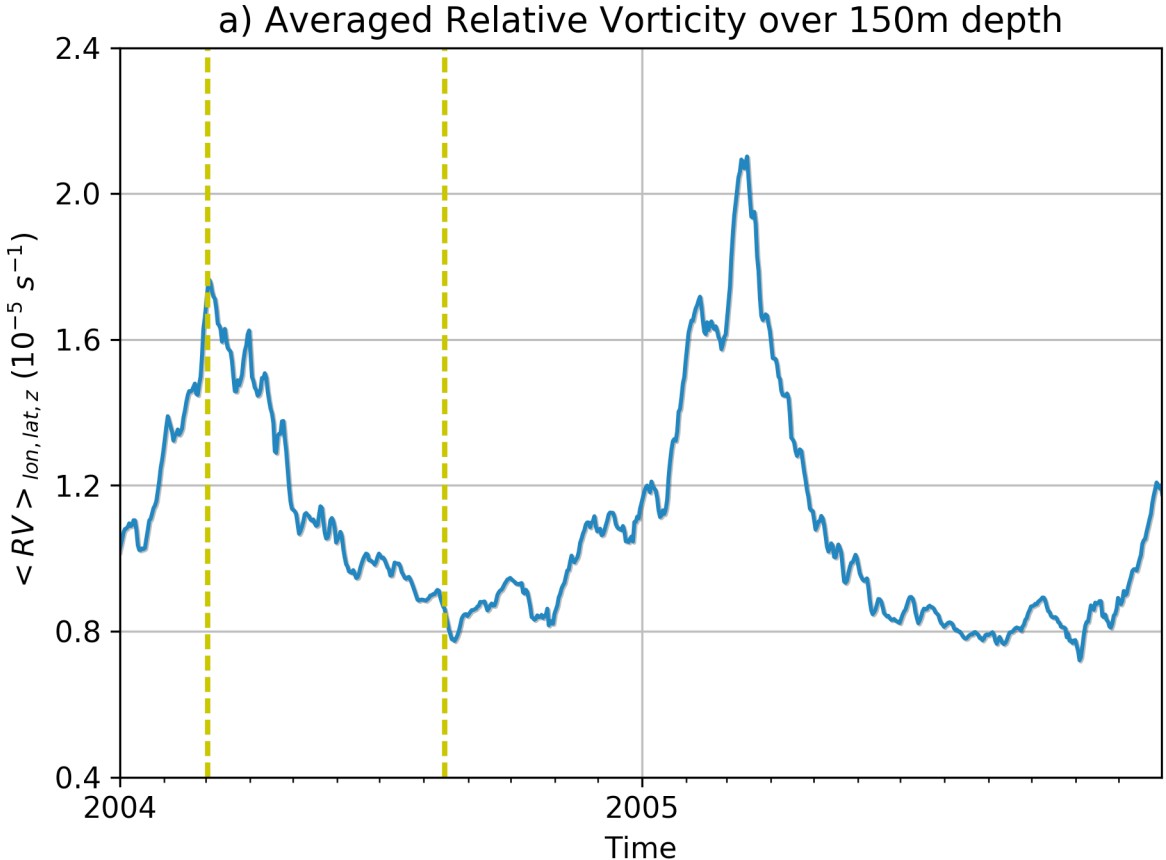

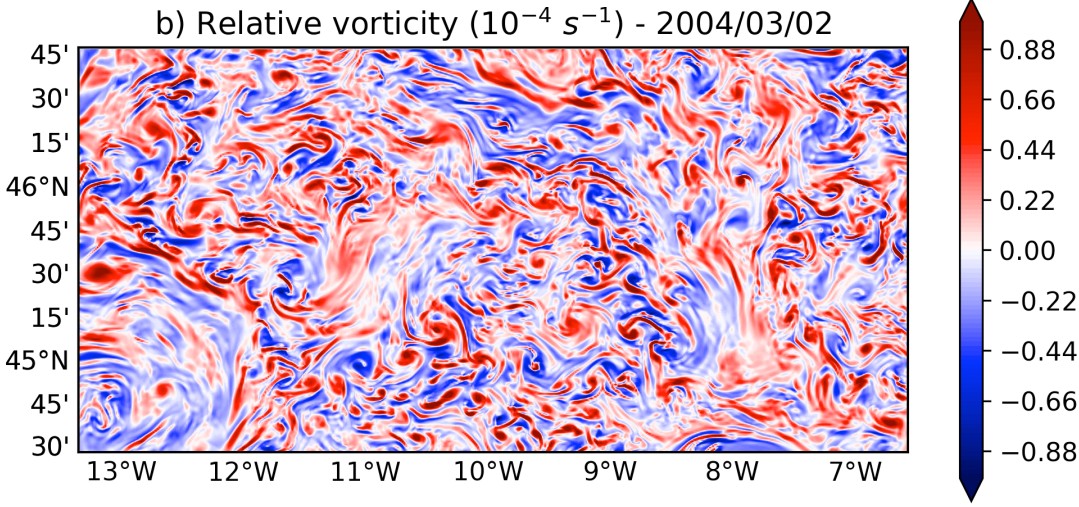

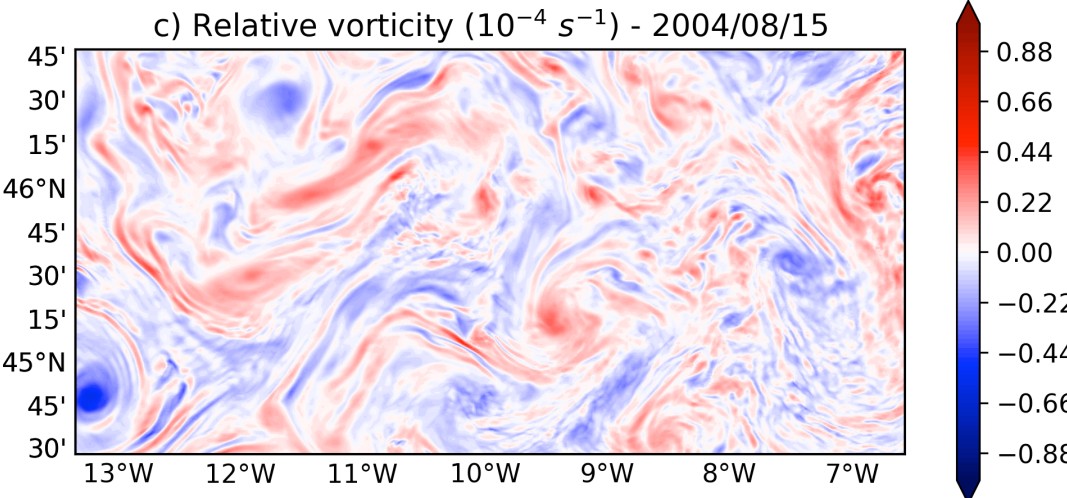

**Figure 10: Relative vorticity averaged over 150m depth and spatially averaged for the years 2004 and 2005 (a). Map of the surface relative vorticity for the 2[nd] March 2004 (b) and the 15[th] August 2004 (c).**

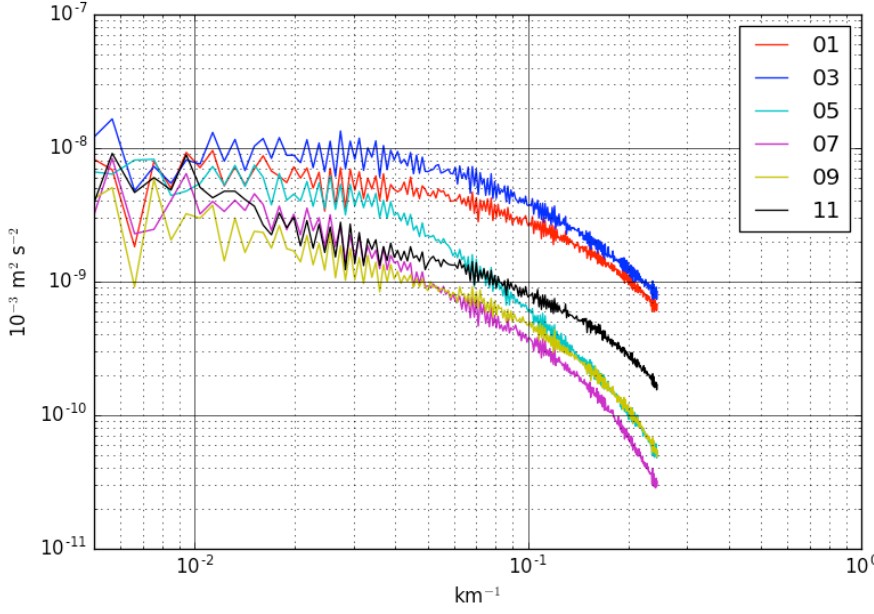

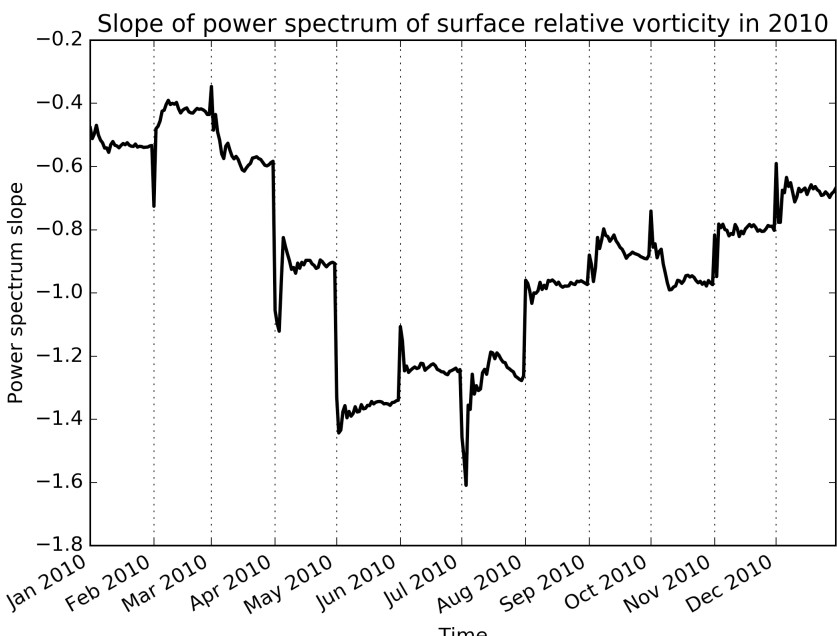

**Figure 11: Power spectrum (computed for each latitude and averaged over longitudes and time during the considered month) from surface relative vorticity for the year 2010 (top). Numbers in legend are corresponding the month in the 2010 year. Time series of the regressed spectral slope from the power spectrum of surface relative vorticity in 2010 (bottom). Spectral slopes have been computed considering wavelengths from 7km to 132 km.**

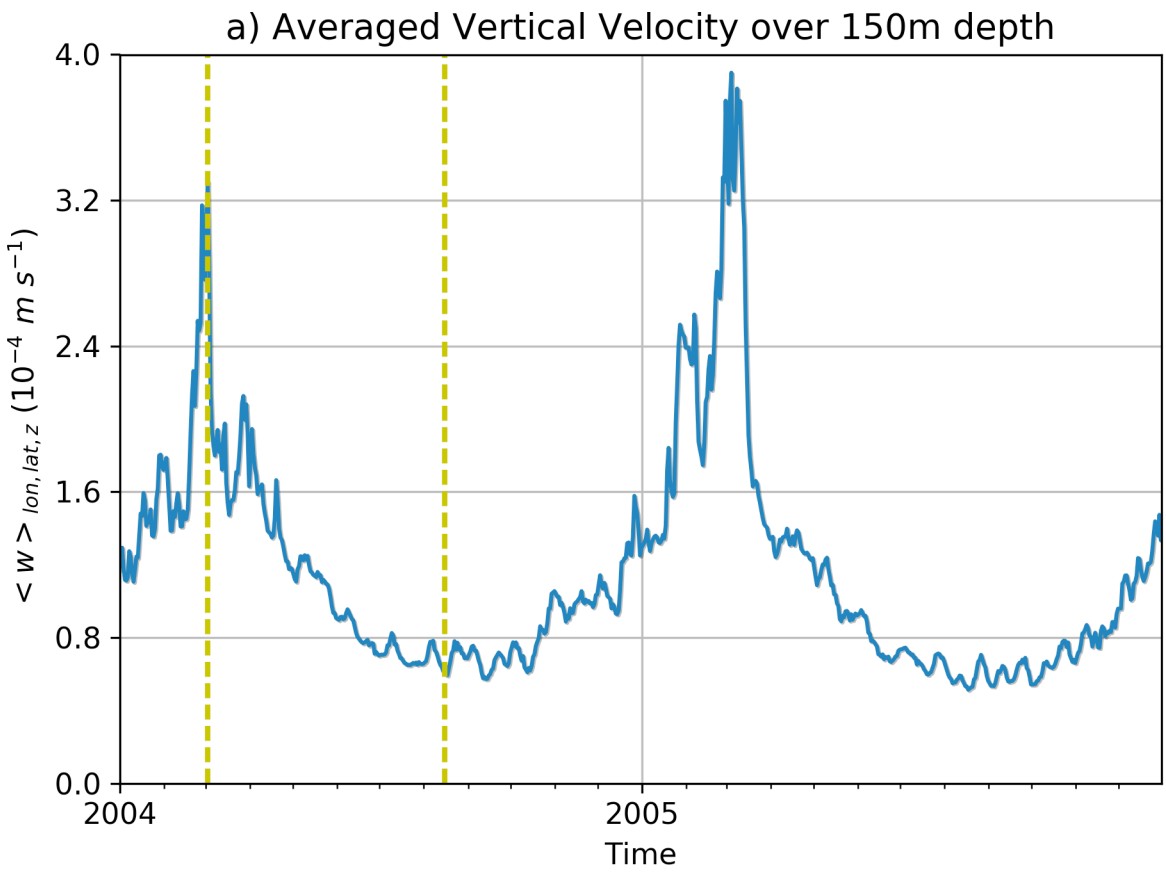

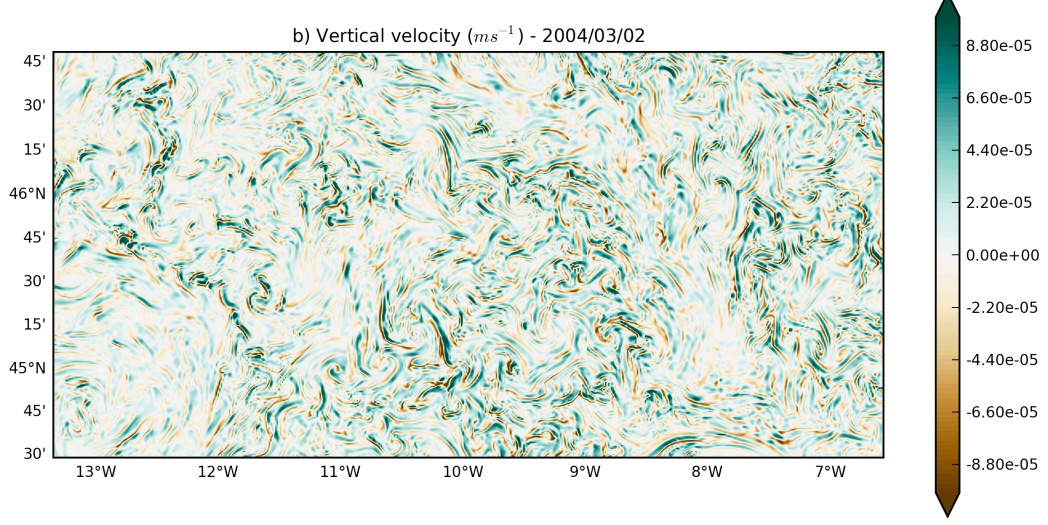

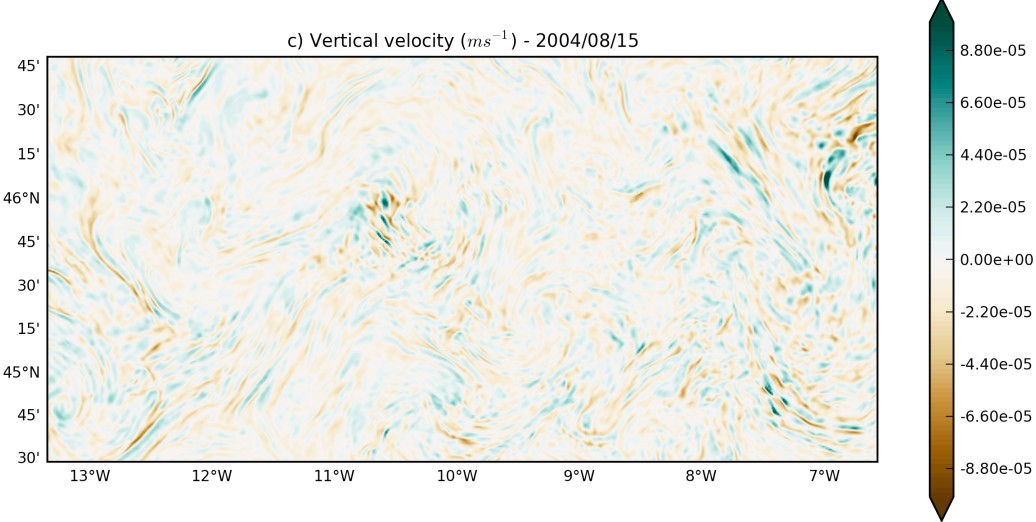

c) Vertical velocity $(ms^{-1})$ - 2004/08/15

**Figure 12: Vertical velocity averaged over 150m depth and spatially averaged for the years 2004 and 2005 (a). Map of the vertical velocity at 4m-depth for the 2<sup>nd</sup> March 2004 (b) and the 15<sup>th</sup> August 2004 (c).**

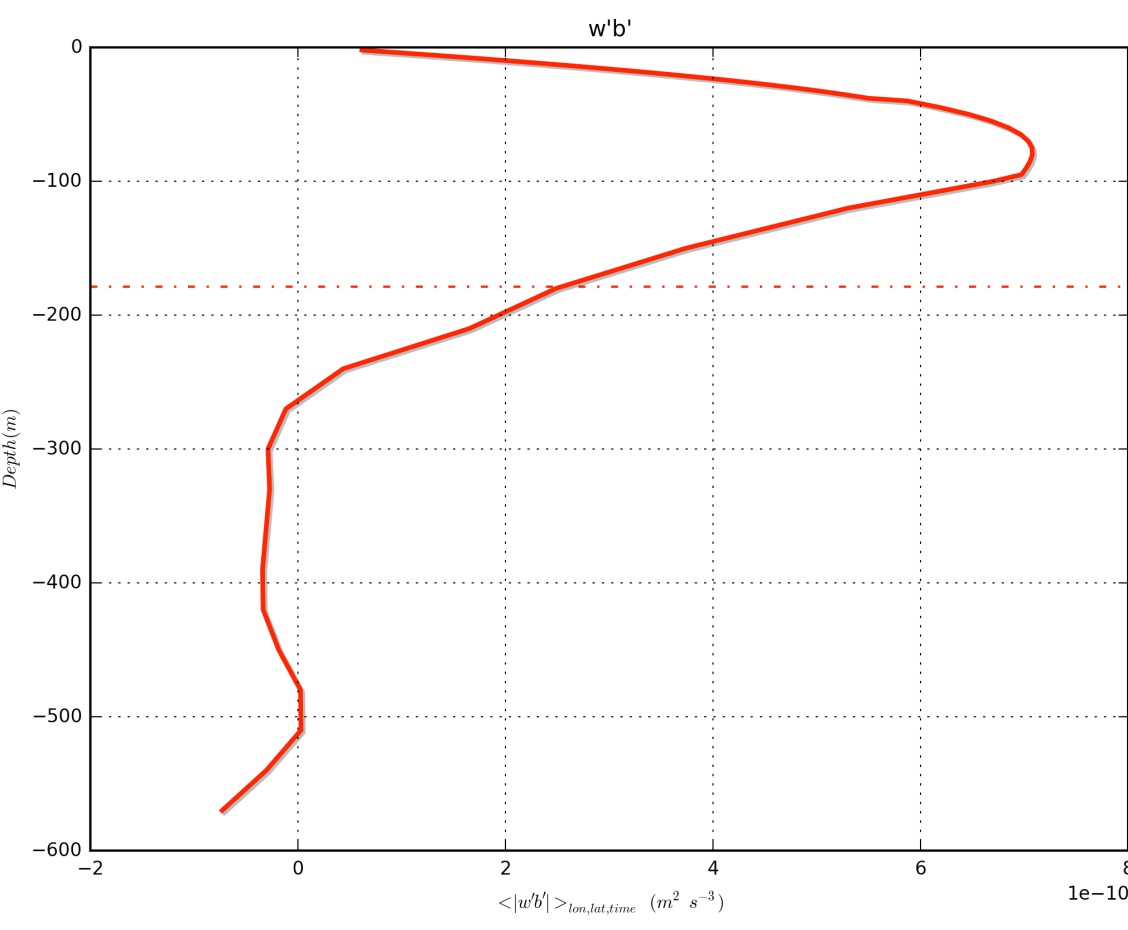

w'b'

**Figure 13: Vertical profile of *w'b'* averaged over the studied subdomain (described in figure 9) during winter season (January-February-March) in 2005. Dashed line represents the mixed layer depth averaged for the same period over the considered region.**

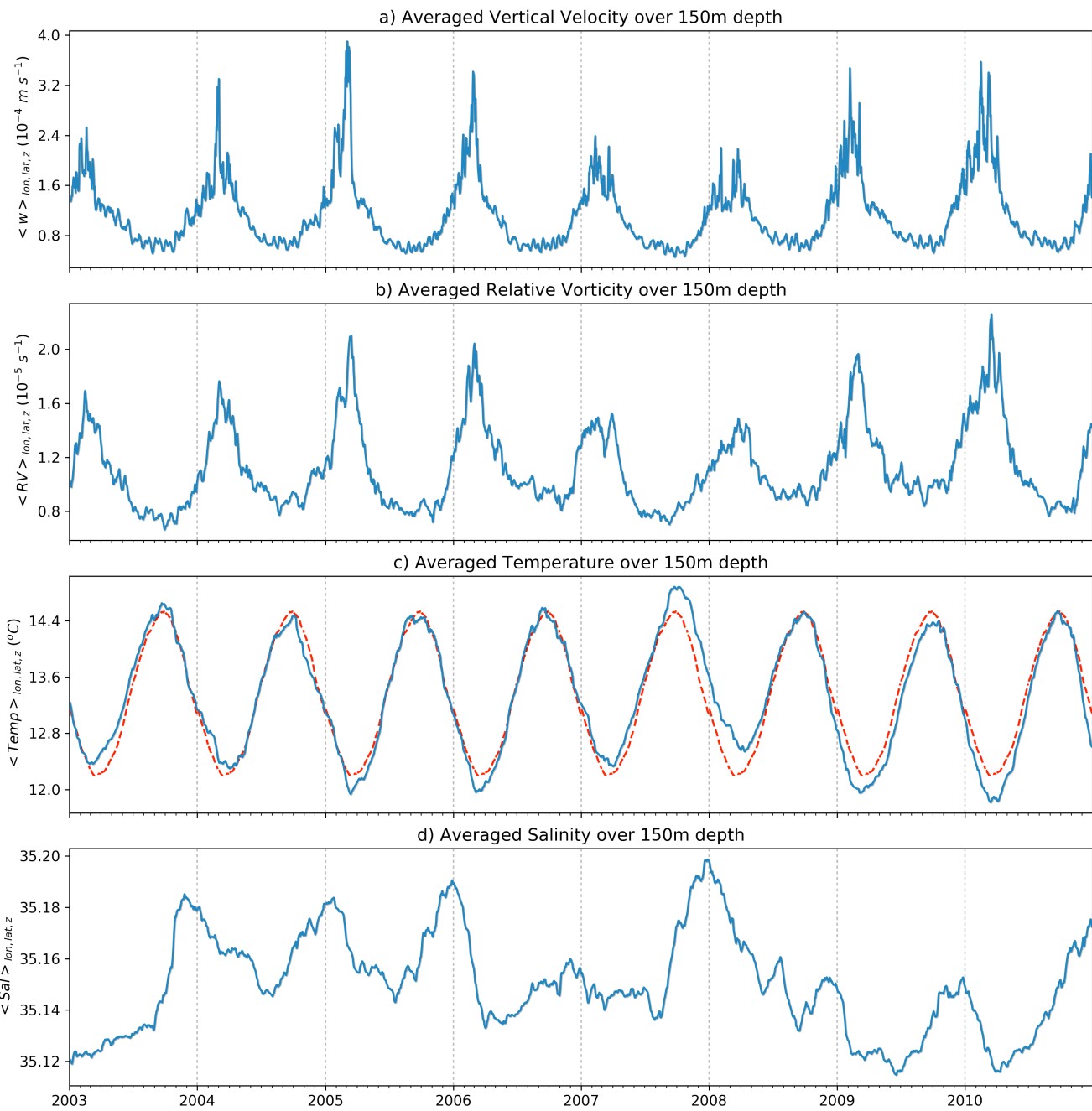

**Figure 14: Interannual variability (from 2003 to 2010) of spatially averaged vertical velocity (a), relative vorticity (b), temperature (the red dashed line is the average annual cycle during the modelled period) (c), and salinity (d) integrated over 150m depth. The considered domain is given in Figure 9.**

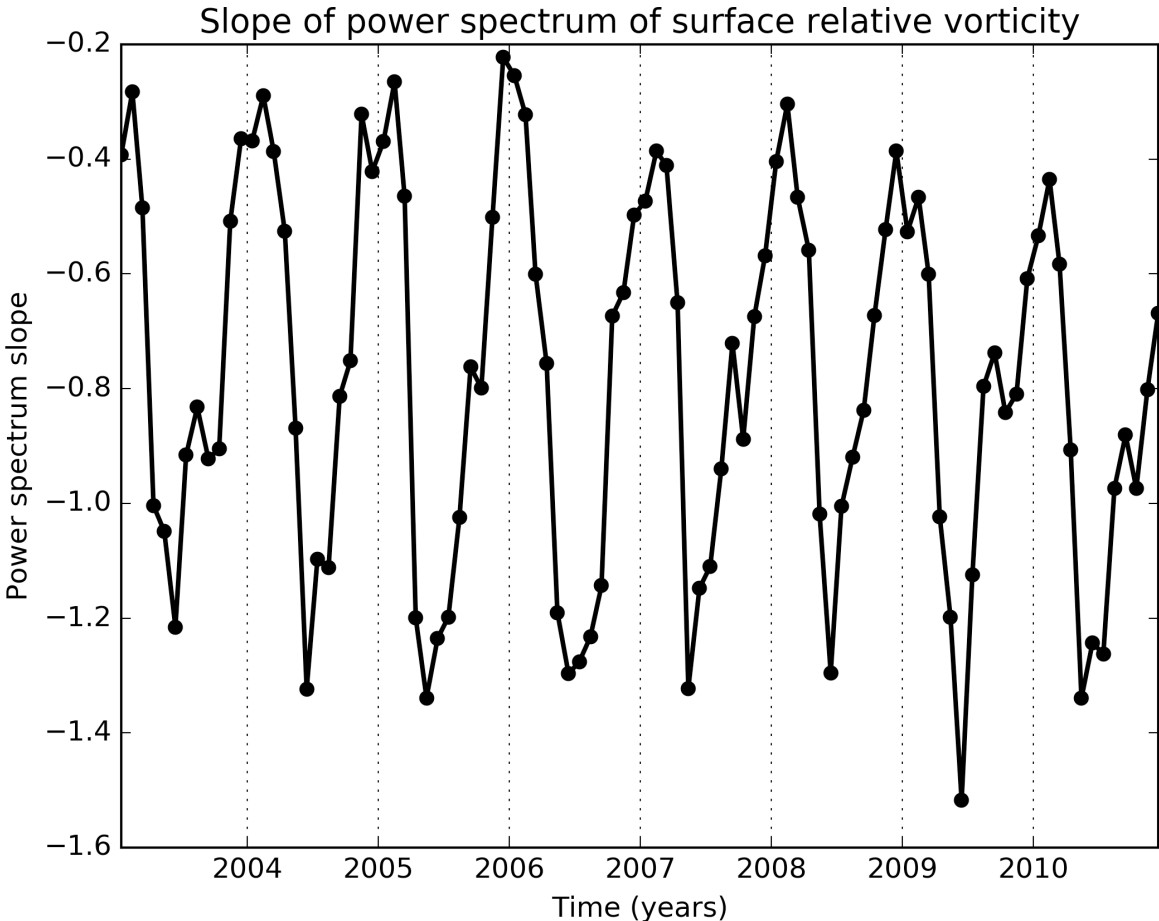

**Figure 15: Time series of the regressed spectral slope from the power spectrum of surface relative vorticity from 2003 to 2010. Spectral slopes have been computed considering wavelengths from 7km to 132 km.**

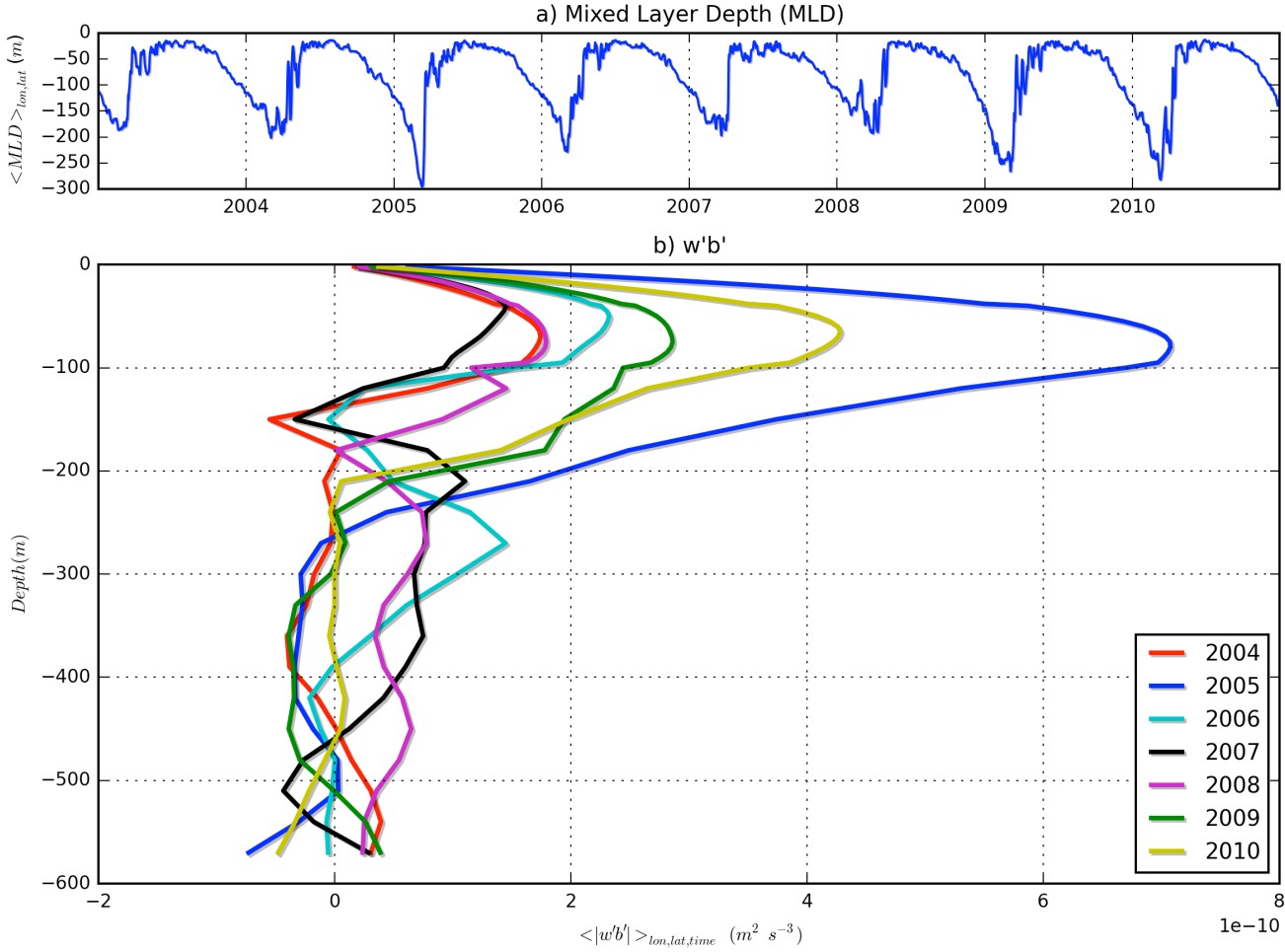

**Figure 16: (a) Averaged Mixed Layer Depth in the studied region (Figure 9). (b) Vertical profiles of w'b' averaged over the same domain during winter seasons (January-February-March).**

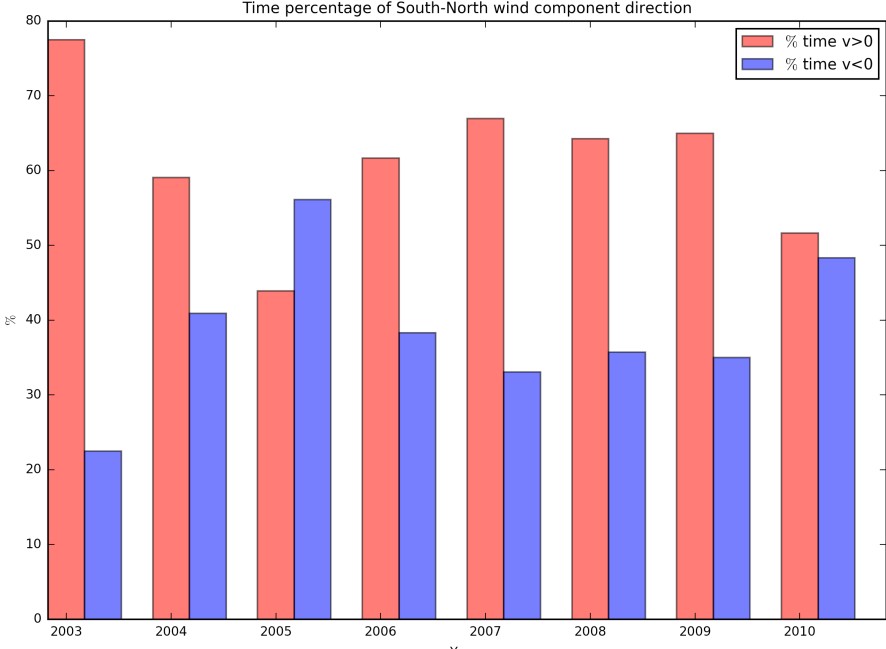

**Figure 17: Statistics on the Northerly and Southerly winds during winters (January-February-March). Based on amotspheric forcings, the percentages of occurrence of Northerly (in blue) and Southerly (in red) winds are represented.**

