# Peer review of "Interannual evolutions of (sub)mesoscale dynamics in the Bay of Biscay"

_Ocean Science, 2016_

## Referee Comment (RC1) · L. Brannigan (Referee) · 1 Mar 2017

This manuscript by Charria et al examines the mesoscale and submesoscale dynamics that emerge from a 1 km resolution simulation of the Bay of Biscay. The authors investigate the extent to which the model provides a realistic representation of the region by comparison with observations. They then consider some metrics of submesoscale dynamics through the seasonal cycle. Finally, the look at the interannual variability of these metrics.

Overall, I think the simulation carried out is an interesting one. The description of the simulation is missing numerous details such as the bottom and lateral boundary conditions - in general the authors should include everything that would be needed for someone else trying to reproduce their experiment. There are various forms of anal-

ysis carried out. However, the description of the analyses are also generally lacking important details. The authors should describe any analysis carried out and set out the equations used to do the analysis. See the line comments below for specific examples of these points.

In my view the analysis in Section 4.1 should be the key result of the paper, but I think that the analysis is relatively limited in this section and the discussion. Much of the previous results on the seasonal cycle are simply reproductions of results found in previous papers that cover other regions and this interannual aspect is really the potentially exciting and novel part of this simulation - as reflected in the title of the manuscript. The authors have shown that there is significant interannual variability in the vorticity, vertical velocity and mixed layer depth fields and that this may be related to surface fluxes. I think they should examine and present figures (probably scatter plots) in the revised draft showing the relationships between the vorticity, vertical velocity and mixed layer depth metrics and the forcing terms. Perhaps simple linear relationships could be investigated between averaged values of buoyancy forcing, wind stress magnitude, down-front winds (see Thomas 2005,http://journals.ametsoc.org/doi/abs/10.1175/JPO2830.1 or Brannigan et al (2015) for an example http://www.sciencedirect.com/science/article/pii/S1463500315000803) as the independent variables with vorticity and vertical velocity as the dependent variables. If the authors want to maintain the length of the paper then some of the more qualitative discussion around Figures 10bc, 12 bc could be removed.

Line-by-line Comments

L22 " The first high resolution" - could be "The first submesoscale-permitting"

L24 - "potentially"

L29 "Our understanding of the general..."

L10 " (adjustment, for example)" - I'm not sure what you're referring to here.

L13 "decreases to values around 5-8 km" - reference for this?

L36 - "the two-dimensional barotropic system is resolved" - I'm not sure what you mean by resolved here.

The equation of state for the simulation, along with bottom and side boundary conditions have not been stated. A comment of how long the simulation takes to spin-up its annual cycle would be useful. Is the model hydrostatic? Given that the model used is not that familiar to many (or me at least) it may be better to set out the continuous form of the model equations.

"$\theta$ and b are surface and bottom control parameters)" - I'm not really sure what these parameters do from this description. Does theta have units? b seems to have units of metres, but they are not mentioned.

L9 "Digital Terrain Model (DTM)"

L13 " are interpolated on the grid and merged" - it's not evident from Figure 1 that the grid resolution is close to 1 km. Perhaps you want to include a second panel that shows a zoomed-in area that highlights the complex topography which the model allows?

L15 - what is h?

L24 - are these horizontal "viscosity/diffusivity values"?

L26 - "is provided"

L29 - "components"

L34 "December 2014"

L37 - are the outputs daily averages or snapshots?

L7 - "over 2010" - to emphasise that the averaging is carried out over the full year.

L9 - "shows a good agreement" - what metric are you using to define what makes for a good agreement?

L14 "Figure 2b"

L14 "the temporal variation of the spatially averaged bias and the associated standard deviation" - this calculation should be given explicitly

L14 "and on average" - average over what?

L18 "The largest"

L23 "In front of the Brittany" - you can't expect all readers to know where Brittany is. A latitude and longitude is much more helpful. This point can be repeated in general for many of the references made to figures.

L26 "and 34.8" - missing units. Indeed salinity units are missing throughout the manuscript.

L31 "The elongated freshwater filaments extending to the Southwest in the Southern" - again, specifying the position of these filaments would be helpful.

L30-35 - this whole paragraph (and the associated figure, see comments below) feels like it has not been developed and does not seem to go anywhere. There is talk of producing some filaments and export, but little or no evidence has been presented for the various statements. If this point on capturing freshwater export is genuinely important then it should be developed, otherwise I'd delete this paragraph.

L37-40 - these statements seem premature given that the vertical structure and flow hasn't been considered yet.

L9 - "Figure 5" L10 "with a small average misfit of 0.015°C" - I'm not sure that the average is the key metric here, because there are lots of offsetting positive and negative errors. The RMS error is also required.

L15 " average misfit is very small at surface" - if we assume the haline coefficient is about 4 times the thermal expansion coefficient, then this misfit is equivalent to a temperature difference of 0.1 degree C. Yet the 0.015 degree thermal difference is

referred to as small, but the effectively 6 times larger salinity difference is referred to as "very small" . This is inconsistent.

L27-35 - the kind of qualitative analysis presented here is subject to "cherry-picking" the parts of the simulation that agree with the observations. Could the model ssh field be compared with altimetry instead? Or (more challenging) could a dynamic height field be constructed from hydrographic observations that can be compared with the observations?

L4 "Two dimensional"

L11 - "A general agreement following the current directions and amplitudes is observed" - can some statistical basis for this be given?

L21 "located on the Aquitaine shelf" - I'd say that few people outside of France knew where this is.

L1 "the relative vorticity" - this needs to be defined explicitly. Presumably you are referring just to the vertical component of relative vorticity and this should be made clear.

L7 "in the northern part of the domain spreading from the shelf break" - give a reference location so the reader knows exactly what you are referring to. "small structures related to local drivers" - same point, there is a very large area and I'm not sure what you're talking about exactly.

L14 "shadowed by large-scale vortices" - do you mean "obscured"? "Shadowed" means more like "followed" in English. Also, I don't think it's physically sound to talk of obscuring by large scale vortices - it implies the small-scale vortices are there, but you just can't see them. In reality, I imagine that the small-scale vortices are simply absent in summer.

L16 "The spatial spectral analysis over the domain" - This analysis would be interesting to see. You could perform an analysis as in Brannigan et al. (2015) whereby you

compute the mean spectral slope over different regions and plot the time series of this slope through the seasonal cycle.

L20 "Based on the spatial average integrated over 150 m depth" - the use of a fixed depth will bias the calculation low in summer time compared to winter. What we're really interested in is the evolution of vorticity in the mixed layer over the year. The calculation show make some effort to track the mean mixed layer through the seasonal cycle, even if it isn't exact.

L22 "The horizontal patterns (Figures 10b and 10c) associated with these average time series confirm the larger range of relative vorticity values related to small scales structures" - you're making a comparison here ("larger") but haven't stated what you're comparing. Using the seasonal spectral slope calculation would allow you to make the seasonal comparison explicitly without having to do this qualitative analysis.

L25 - " surface relative vorticity spectra" - you need to set out explicitly how you did this calculation. You should also justify why you choose to use the vertical component of relative vorticity for this rather than a kinetic energy spectrum. The spatial pattern used to calculate the spectra is also unclear.

L26 " seasonal variation of the energy" - presumably you mean the variance here, as the underlying spectra is of vorticity.

L30 "The relative vorticity fields, related to vortices" - this isn't right, the vorticity comes from a mix of vortices, jets and filaments.

L30 "The relative vorticity fields, related to vortices, can also be explored through vertical motions. " - This sentence doesn't make sense to me. Surely we are investigating the overall model dynamics, and the vorticity and vertical motions are two separate diagnostics for this?

L30 " The role of these structures" - again, this sentence implies that vortices are the only processes of interest, when it is a mix of dynamically distinct processes.

L6 "The diagnostic"

L15 - this comment on the spin-up period should be in the second section. Some justification needs to be given as to why the model is considered to have spun-up in two years.

L15 - I think it would be more intuitive to divide the values in Figure 14 by 150 m to get mean values, I've no idea how to judge the magnitude of vertical velocity in units of mˆ2/sˆ-1

L26 "A model simulation...exhibits"

L28 "theatre" change to "location"

L34 " and to the seasonal restratification" - this hasn't been shown in the paper. In any event, the seasonal restratification is likely to be driven primarily by heating at the surface. This is shown by the fact that a 1D model in the vertical produces adequate seasonal restratification without any MLIs.

L36 " Indeed, Soufflet et al (2016)" - this isn't included in the list of references.

A number of figures are missing x, y and colorbar axis labels. The text on the labels is also generally quite small and hard to read. Maybe it could be made into a semi-bold font?

Figure 1 - the non-uniform color scheme here is not a good choice - it strongly highlights the yellow region which may or may not be the region of strongest gradients. A uniform perception scheme is recommended instead.

Figure 2 - the title should be in symbols rather than the computer code format used. This is confusing as it's not clear to me if the < > are being used as mathematical operators here or just part of your text formatting. The temporal averaging that is required to make the calculation in the plot should be made explicit in the text. "The shape of the curves represents the spatial standard deviation" - this seems incorrect,

the shape of the curves is the seasonal cycles, but it seems like you're actually talking about the shading around the curves.

Figure 4 - why are there two subplots? Why is one of them smaller than the other? Does the second one even get referred to?

Figure 6 - shouldn't these plots show the average over 50 grid points rather than just selecting one grid point out of 50?

Figure 8 - a red-white-blue colorbar should be used instead of a rainbow color schemes. In general rainbow color schemes are known to distort perception of data and should be avoided. The figure titles could mention which is the model and which is the observation.

Figure 9 - it is more helpful to plot the relative vorticity normalised by the local value of f (and other vorticity plots).

Figure 12 - the depth of the plan views needs to be stated.

---

## Referee Comment (RC2) · Anonymous Referee #2 · 13 Mar 2017

This is a relatively short paper documenting the seasonal variability in a high resolution, regional numerical model of the upper ocean in the Bay of Biscay. It is first demonstrated that the model faithfully reproduces the large space and time scale variations in the region. The authors then go on to describe the level of submesoscale variability through the spectral energy of vorticity and vertical buoyancy flux. A relationship between the depth of winter mixing and the energy in the submesoscale is observed.

This paper is well written and the model seems to provide a useful representation of the meso- to submesoscale variability in this region. However, aside from documenting the realism of this particular model, I do not find anything new or novel in the paper. Theories exists that relate submesoscale energy to mixed layer depth (among other things), so the present qualitative finding is entirely expected. It is stated that their

results show the importance of submesoscale activity, but this is really only implied. It could be demonstrated by comparison with an otherwise identical model that did not resolve the submesoscale, but this is not done. First one would have to define what quantity they were interested in. It is possible that the submesoscale is important for some things and not for others. The title is somewhat misleading since the submesoscale is discussed in only about 1 1/2 pages and the interannual in only one short paragraph.

I see this as an editorial decision. I did not really learn much from reading the paper, but it does fairly clearly document some aspects of the fidelity of this regional model. If that fits the goals of the journal then the paper could probably be suitable for publication with some revisions. However, my own recommendation would be to reject the paper since I do not see any reasonable revisions leading to new insights. That is not to say that the model does not contain new and interesting things that could be explored and understood, it is just that does not seem to be the authors objective for writing this paper.

Detailed comments:

(Page 2, line 8): There really isn't any connection in the paper between submesoscale activity and climate change.

(3,23) More details are needed on the lateral boundary conditions. Is the sponge layer just a region of high viscosity or are the model prognostic variables are restored towards the ORCA12 variables? Are the ORCA12 variables interpolated and imposed on the boundaries? If so, are the tidal components then added? Just velocity or do the tides perturb the density field as well? Is anything done to sea surface height? What is the temporal resolution of the ORCA12 data?

(3,26) Is there no restoring for salinity, just a surface flux boundary condition?

(4,15) It would be helpful to indicate the annual mean values for the model and observations on the figure.

(4,21) It would be helpful here and elsewhere to mark on the figures various geographic features described in the text to help orient the reader.

Section 3.2: What is the standard deviation of the error, and what is the standard deviation in each the model and observations. Some of this spread is due to eddies and different phasing but we can't tell if the model is getting the statistics of the eddies correct or not.

(5,26) These vectors are very difficult to see. Maybe make the arrowheads bigger. In general the figures need to use larger fonts, they are very difficult to read.

(6,5) The agreement suggests that at least some of this variability is forced, not internal.

(6,23) Can the observations be included here? No one is going to track down that paper.

(7,5) It would be helpful to include at least one topographic contour so we can tell where the ocean transitions from shallow to deep.

(7,8) How do you know these features are related to local drivers?

(7,23) How does the high winter energy decay into the low spring energy? Is it dissipated locally or radiated away?

(7,30) The discussion implies that vertical velocity and mixing are directly related but one can have very large vertical velocities through baroclinic instability and no diapycnal mixing.

(8,21) It looks like there is some energy in salinity at the seasonal period from Fig. 14.

(8,32) It is not clear what is meant by instabilities driving potential energy to dissipation.

(10,5) The interannual variability was not the focus of the paper. The paper really focusses on validating the model.

There were also many minor grammatical errors and unclear phrasing, but given my larger concerns about the direction of the paper I have not detailed these issues here.

---

## Author Comment (AC1) · 21 May 2017

This manuscript by Charria et al examines the mesoscale and submesoscale dynamics that emerge from a 1 km resolution simulation of the Bay of Biscay. The authors investigate the extent to which the model provides a realistic representation of the region by comparison with observations. They then consider some metrics of submesoscale dynamics through the seasonal cycle. Finally, the look at the interannual variability of these metrics.

Overall, I think the simulation carried out is an interesting one. The description of the simulation is missing numerous details such as the bottom and lateral boundary conditions - in general the authors should include everything that would be needed for someone else trying to reproduce their experiment. There are various forms of analysis carried out. However, the description of the analyses are also generally lacking important details. The authors should describe any analysis carried out and set out the equations used to do the analysis. See the line comments below for specific examples of these points.

In my view the analysis in Section 4.1 should be the key result of the paper, but I think that the analysis is relatively limited in this section and the discussion. Much of the previous results on the seasonal cycle are simply reproductions of results found in previous papers that cover other regions and this interannual aspect is really the potentially exciting and novel part of this simulation - as reflected in the title of the manuscript. The authors have shown that there is significant interannual variability in the vorticity, vertical velocity and mixed layer depth fields and that this may be related to surface fluxes. I think they should examine and present figures (probably scatter plots) in the revised draft showing the relationships between the vorticity, vertical velocity and mixed layer depth metrics and the forcing terms. Perhaps simple linear relationships could be investigated between averaged values of buoyancy forcing, wind stress magnitude, down-front winds (see Thomas 2005,http://journals.ametsoc.org/doi/abs/10.1175/JPO2830.1 or Brannigan et al (2015) for an example http://www.sciencedirect.com/science/article/pii/S1463500315000803) as the independent variables with vorticity and vertical velocity as the dependent variables. If the authors want to maintain the length of the paper then some of the more qualitative discussion around Figures 10bc, 12 bc could be removed.

Following one of the major referee comments, the discussion has been modified including more emphasized discussion on the interannual variability. The relationship between forcing terms and ocean features has been discussed including new details on the process explained. However, this remains more limited than expected as we were facing a technical problem to give a full detailed description of the turbulent fluxes (latent and sensitive fluxes) as they have not been saved for this simulation and we do not have the available computing time and infrastructure to perform the same simulation with more saved variables. Nevertheless, we explored the different possible improvements based on offline diagnostics and we propose more detailed analyses to confirm our proposed assumptions in the discussion.

Compared with the previous manuscript, a discussion has also been added on the interannual evolution of the spectral content based on spectral slope analysis (as suggested by the referee).

**Line-by-line Comments**

L22"Thefirsthighresolution"-couldbe"Thefirstsubmesoscale-permitting"
Following referee suggestion, the text has been replaced.

L24 - "potentially"
The word has been corrected.

L29 "Our understanding of the general..."
The sentence has been improved following suggestion.

L10 " (adjustment, for example)" - I'm not sure what you're referring to here.
In this sentence, we just mention the role of the water depth in the definition of the Rossby radius.
The sentence has been replaced by: "... The considered definition for the studied scales, depending on the depth of the water column and the stratification, has to be recalled as we progress in a coastal environment. ..."

L13 "decreases to values around 5-8 km" - reference for this?
This values are coming from an internal report (in French - *Valdivieso Da Costa M., Maze J.-P., Raynaud S., Etude de plan d'échantillonnage de PAGODE sur le plateau du golfe de Gascogne - Lot 1 – Analyse de la variabilité interannuelle de la température à la surface de la mer et des rayons de déformations internes - Référence ACTIMAR : 06.34.1, 2006*) not available online. Then, there is no reference for these values. A publication on this point is under preparation.

[Figure]

Upper-left: First mode of the internal Rossby radius for the whole CTD dataset (6448 profiles from 1972 to 2005 from SHOM and IFREMER). Other figures are representing the same parameter for a different range of depths and adapted color scales.

To improve our manuscript, the following statement has been added:

5  "Over the continental shelf, this internal Rossby radius of deformation decreases to values around 3-8 km (Valdivieso Da Costa M., Maze J.-P., Raynaud S., pers. comm., 2006), for example, in the Bay of Biscay."

L36 - "the two-dimensional barotropic system is resolved" - I'm not sure what you mean by resolved here.
The text has been modified as the "two-dimensional barotropic system" is the barotropic mode.
10

The equation of state for the simulation, along with bottom and side boundary conditions have not been stated. A comment of how long the simulation takes to spin-up its annual cycle would be useful. Is the model hydrostatic? Given that the model used is not that familiar to many (or me at least) it may be better to set out the continuous form of the model equations.
"θ and b are surface and bottom control parameters)" - I'm not really sure what these parameters do from this description.
15  Does theta have units? b seems to have units of metres, but they are not mentioned.
Indeed, the statement was not clear enough. First, there was a mistake in the reference (corrected in the new manuscript). It was Lazure and Dumas (2008) as given in the reference list and not Lazure et al. (2005). All the equations are detailed in this publication (Lazure and Dumas, 2008). However, for clarity purpose, equations have been added in an appendix of the paper and details are given.
20  The model is hydrostatic.
Concerning the spin-up, its length is mentioned in section 4.1 (as commented by the referee) and taken to 2 years. Following the different experiments to develop this configuration and to produce this simulation, this 2-year period has been defined based on the development of the seasonal cycle (for example in temperature). To support this choice, we explored the simulation in terms of circulation and temperature. The Bay of Biscay is following two main regimes. One over the
25  continental shelf where the circulation is strongly driven by atmospheric forcings and river inputs, and then is balance after few months. The other one is over the deep ocean where the large-scale circulation is constrained through open boundary conditions. As the domain is limited, the 2-year period is a long enough period (usually, we consider even shorter spin-up for this region).
In the manuscript, this spin-up period is mentioned in the section 2.2 as:
30  " ... until December, 31[st], 2010. For interannual analyses, a spin-up of two years is taken into account to setup an established seasonal cycle in the circulation even in the open ocean constrained by large-scale solution forced in open boundary conditions. The analysed period is then running from 2003 to 2010. The BACH1000_100lev configuration is implemented ..."

35  Concerning the sigma-coordinates parameters ($h_c$=200 m, θ=6 and b=0), following referee suggestions, they have been detailed in the Appendix A with the model equations.

L9 "Digital Terrain Model (DTM)"
The acronym has been added.
40

L13 " are interpolated on the grid and merged" - it's not evident from Figure 1 that the grid resolution is close to 1 km. Perhaps you want to include a second panel that shows a zoomed-in area that highlights the complex topography which the model allows?
In order to minimize pressure gradient errors due to sigma coordinates, the bathymetry remains smoothed in the simulation.
45  The used filter is based on the criteria defined in Haidvogel and Beckmann (1999) and usually used for the ROMS model (as described in the manuscript - Numerical experiments section). However, the MARS3D model allows keeping a fine resolution for the coastline. A zoomed-in area has been added in the manuscript (Figure 1) around 48°N to illustrate this resolution.
Furthermore, following referee sugestions, the colormap has been adjusted to a colormap with a uniform perception scheme.
50

L15 - what is h?

h is the depth of the water column. It has been added in the manuscript.

L24 - are these horizontal "viscosity/diffusivity values"?
Indeed, viscosity/diffusivity values are horizontal.
The manuscript has been modified as:
"The sponge layer width is 20 km-width and the maximum horizontal viscosity/diffusivity values are 100 m$^2$ s$^{-1}$ and zero outside the open boundary layers."

L26 - "is provided"
The sentence has been corrected.

L29 - "components"
It has been corrected.

L34 "December 2014"
It has been corrected.

L37 - are the outputs daily averages or snapshots?
Outputs are daily averages. The manuscript has been modified to mention this point.

L7 - "over 2010" - to emphasise that the averaging is carried out over the full year.
The manuscript has been updated following referee suggestion.

L9 - "shows a good agreement" - what metric are you using to define what makes for a good agreement?
Following referee suggestion, the sentence has been modified to include more detailed metrics to quantify the difference between modelled and observed sea surface temperature fields.
The updated manuscript is including these new sentences:
" ... In Figure 2a, the mean bias between observed and modelled SST over 2010 at the end of the experiment shows an averaged underestimation of the temperature by the model (-0.24±0.28°C over the domain). These biases do not exceed 1.75 °C. Such large values are obtained in two regions. First, in the Ushant ..."

L14 "Figure 2b"
L14 "the temporal variation of the spatially averaged bias and the associated standard deviation" - this calculation should be given explicitly
L14 "and on average" - average over what?
Following referee remark, the calculation has been introduced explicitly in the manuscript.
The average is done over the full domain. The sentence has been modified: "... scale and on average over the full domain ...".

L18 "The largest"
It has been corrected.

L23 "In front of the Brittany" - you can't expect all readers to know where Brittany is. A latitude and longitude is much more helpful. This point can be repeated in general for many of the references made to figures.
The positions of described regions are explicitly given in the manuscript.
For example:
"... In front of the Brittany (48.2°N / 5.6°W), the position ..."
"... in the Southern part of the Bay of Biscay (44°N / 3.5°W) are representing ..."

L26 "and 34.8" - missing units. Indeed salinity units are missing throughout the manuscript.
Indeed, no units for salinity have been used. However, as stated in UNESCO technical report (1985, p. 44 for salinity; Millero et al., 2008) and the international system of units, the Practical Salinity units, defined as conductivity ratio, is "dimensionless". Following this definition, we propose to keep the no unit convention in the paper.

- UNESCO (1985) The international system of units (SI) in oceanography, UNESCO Technical Papers No. 45, IAPSO Pub. Sci. No. 32, Paris, France.
- Millero, F. J., Feistel, R., Wright, D. G., & McDougall, T. J. (2008). The composition of Standard Seawater and the definition of the Reference-Composition Salinity Scale. *Deep Sea Research Part I: Oceanographic Research Papers*, *55*(1), 50-72.

Example of recent paper following this convention:
Boutin, J., Martin, N., Reverdin, G., Yin, X., and Gaillard, F.: Sea surface freshening inferred from SMOS and ARGO salinity: impact of rain, Ocean Sci., 9, 183-192, doi:10.5194/os-9-183-2013, 2013.

L31 "The elongated freshwater filaments extending to the Southwest in the Southern" - again, specifying the position of these filaments would be helpful.
The position has been written in the manuscript: "... in the Southern part of the Bay of Biscay (44°N / 3.5°W) are representing ..."

L30-35 - this whole paragraph (and the associated figure, see comments below) feels like it has not been developed and does not seem to go anywhere. There is talk of producing some filaments and export, but little or no evidence has been presented for the various statements. If this point on capturing freshwater export is genuinely important then it should be developed, otherwise I'd delete this paragraph.
We understand the referee point of view on this paragraph. However, we decided to keep it, as it remains a challenging issue in the region (modelling cross-shelf exchanges in the Bay of Biscay). In the discussed paper (Reverdin et al., 2013), coastal model with coarser resolution were not able to reproduce this event. Presenting in this publication the ability of the model in this configuration to reproduce this event is interesting for future studies. Developing this part could be interesting but it could lead us far beyond the scope of the paper, then we kept a quick overview on this event.

L37-40 - these statements seem premature given that the vertical structure and flow hasn't been considered yet.
Indeed, this statement is not at the right place and it has been removed.

L9 - "Figure 5" L10 "with a small average misfit of 0.015°C" - I'm not sure that the average is the key metric here, because there are lots of offsetting positive and negative errors. The RMS error is also required.
We agree with the referee and we included the RMS error in this part of the manuscript. The RMS error is giving another view of the model/observation differences and has to be considered in the frame of a configuration with several assumptions. This last point has been given in the manuscript.

L15 " average misfit is very small at surface" - if we assume the haline coefficient is about 4 times the thermal expansion coefficient, then this misfit is equivalent to a temperature difference of 0.1 degree C. Yet the 0.015 degree thermal difference is referred to as small, but the effectively 6 times larger salinity difference is referred to as "very small" . This is inconsistent.
The manuscript was, indeed, not consistent on this point. The sentence has then been modified following referee suggestion.
The new sentence is "... In salinity, the average misfit is smaller at surface (-0.024 for the layer 0-20 m depth, RMSE=0.27, Figure 5d) and above 100 m depth (0.025 for the layer 40-100 m depth, RMSE=0.28 Figure 5e and 5f), than between 20 and 40 m depth where the average difference is larger (0.176, RMSE=0.59). ... ".

L27-35 - the kind of qualitative analysis presented here is subject to "cherry-picking" the parts of the simulation that agree with the observations. Could the model ssh field be compared with altimetry instead? Or (more challenging) could a dynamic height field be constructed from hydrographic observations that can be compared with the observations?
The choice of the comparisons can seems like " cherry-picking the parts of the simulation that agree with the observations" but they are more related with the availability of observations. At the end, we agree that it can be seen as a patchwork of validations but it is observation driven for the region and the period.
The possible comparison with SSH has been explored (and remains under investigation in the framework of the future SWOT satellite) but we faced several issues as the quality of the Mean Dynamic Topography in the region, the macro-tidal system over the continental shelf (tide are tricky to correct in the region). Then, it has been decided to not include these figures in the manuscript.
For the dynamic height, the problem is the representativeness of in situ observations as they are limited to explore interannual variability and then, the climatology can be biased due to the difference of observation distribution depending the year.
These different points explain the choice of keeping the averaged current map to validate the general circulation in the region.

L4 "Two dimensional"
The text has been modified.

L11 - "A general agreement following the current directions and amplitudes is observed." - can some statistical basis for this be given ?
Some statistics have been added in the manuscript.
Please find below, the detail of those statistics:
ASPEX 4
Root Mean Square Error :
cross-shore = 0.0244893 m s$^{-1}$
along-shore = 0.0378377 m s$^{-1}$

Correlation :
cross-shore = 0.36939642463
along-shore = 0.605244276335

Standard deviation :
cross-shore BACH1000 =  0.0134921 m s$^{-1}$
along-shore BACH1000 =  0.0349041 m s$^{-1}$
cross-shore ASPEX =  0.0260202 m s$^{-1}$
along-shore ASPEX =  0.0467999 m s$^{-1}$

#################
ASPEX  10
Root Mean Square Error :
cross-shore =  0.0223718 m s$^{-1}$
along-shore =  0.0639827 m s$^{-1}$
Correlation :
cross-shore =  0.0910204076364
along-shore =  0.68939054688

Standard deviation :
cross-shore BACH1000 =  0.0175352 m s$^{-1}$
along-shore BACH1000 =  0.0862604 m s$^{-1}$
cross-shore ASPEX =  0.0138124 m s$^{-1}$
along-shore ASPEX =  0.073085 m s$^{-1}$

L21 "located on the Aquitaine shelf" - I'd say that few people outside of France knew where this is.
Indeed, we agree that the Aquitaine shelf is not very famous. However, in this case, we refer to Figure 1 where the ADCP position is explicitly given.
To clarify this point, the sentence has been modified as:
"... This mooring located on the Aquitaine shelf (South of 45°N, Figure 1) has been used ..."

L1 "the relative vorticity" - this needs to be defined explicitly. Presumably you are referring just to the vertical component of relative vorticity and this should be made clear.
Indeed, the "relative vorticity in this study corresponds to the vertical component of the vertical vorticity. The following sentence has been modified:
"... the vertical component of the relative vorticity (referred as relative vorticity) has been first analysed. ..."

L7 "in the northern part of the domain spreading from the shelf break" - give a reference location so the reader knows exactly what you are referring to. "small structures related to local drivers" - same point, there is a very large area and I'm not sure what you're talking about exactly.
Following referee suggestion, considered area have been detailed.
The manuscript is then modified as:
"... from the shelf break (around 47.5°N-48°N / 7°W-5°W). In ..."
This sentence has been added to explain more which region is considered: " These structures can be seen through large relative vorticity values over the outer part of the continental shelf between the 100m isobath (Figure 6) and the shelf break (Figure 1). "
Indeed, it covers a large area (like an alongshore band) between 100m isobath (Figure 6) and the shelf break.

L14 "shadowed by large-scale vortices" - do you mean "obscured"? "Shadowed" means more like "followed" in English. Also, I don't think it's physically sound to talk of obscuring by large scale vortices - it implies the small-scale vortices are there, but you just can't see them. In reality, I imagine that the small-scale vortices are simply absent in summer.
We agree with the referee that our sentence was misleading. We then change the text as follows: "... fully developed in winter. In summer, typical spatial scales are larger in than in winter. More large scales vortices are simulated during this season. The spatial ..."

L16 "The spatial spectral analysis over the domain" - This analysis would be interesting to see. You could perform an analysis as in Brannigan et al. (2015) whereby you compute the mean spectral slope over different regions and plot the time series of this slope through the seasonal cycle.
This sentence was confusing as is has been done in Figure 11 (referred now in the manuscript). However, the spectral slope is an interesting indicator to track the seasonal cycle of the energy content. We added the slope time series in Figure 11 (bottom) for the year 2010. The manuscript has been updated.

L20 "Based on the spatial average integrated over 150 m depth" - the use of a fixed depth will bias the calculation low in summer time compared to winter. What we're really interested in is the evolution of vorticity in the mixed layer over the year. The calculation show make some effort to track the mean mixed layer through the seasonal cycle, even if it isn't exact.
We agree with the referee that the choice of a fixed depth is an important assumption for the study. However, the idea of the study was to track the behaviour of the surface layer that can be fully mixed in winter. To analyse the time series and to

compare different season, we need to take into account the same water depth to infer the activity in this surface layer. However, using the changing mixed layer depth as the depth of averaging is an interesting approach to track the dynamics inside this dynamic layer. We will consider this approach in future analyses.

L22 "The horizontal patterns (Figures 10b and 10c) associated with these average time series confirm the larger range of relative vorticity values related to small scales structures" - you're making a comparison here ("larger") but haven't stated what you're comparing. Using the seasonal spectral slope calculation would allow you to make the seasonal comparison explicitly without having to do this qualitative analysis.
In Figure 11, the spectral slope has been added and used to support the explanation in the manuscript.

L25 - " surface relative vorticity spectra" - you need to set out explicitly how you did this calculation. You should also justify why you choose to use the vertical component of relative vorticity for this rather than a kinetic energy spectrum. The spatial pattern used to calculate the spectra is also unclear.
This sentence has been added to explain the spectrum calculation:
"These spectra have been computed every day over the limited domain (yellow rectangle Figure 9) using a 2D Fast Fourier Transform and then averaged over the considered month."
The computation could have been done from kinetic energy (as suggested by the referee) but, as we illustrated and discussed the vertical component of the surface relative vorticity, it makes more sense to do spectrum on this field as it is related with the kinetic energy spectrum.

L26 " seasonal variation of the energy" - presumably you mean the variance here, as the underlying spectra is of vorticity.
The text has been modified following referee suggestion.

L30 "The relative vorticity fields, related to vortices" - this isn't right, the vorticity comes from a mix of vortices, jets and filaments.
L30 "The relative vorticity fields, related to vortices, can also be explored through vertical motions. " - This sentence doesn't make sense to me. Surely we are investigating the overall model dynamics, and the vorticity and vertical motions are two separate diagnostics for this?
L30 " The role of these structures" - again, this sentence implies that vortices are the only processes of interest, when it is a mix of dynamically distinct processes.
We fully agree with the referee and we went back to the manuscript to be more precise and avoid this wrong shortcut between relative vorticity and vortices.

L6 "The diagnostic"
The text has been corrected.

L15 - this comment on the spin-up period should be in the second section. Some justification needs to be given as to why the model is considered to have spun-up in two years.
This point has been discussed in the second section and details are given to the referee above.

L15 - I think it would be more intuitive to divide the values in Figure 14 by 150 m to get mean values, I've no idea how to judge the magnitude of vertical velocity in units of mˆ2/sˆ-1
Following referee suggestion, Figure 14 and Figure 9 have been updated.

L26 "A model simulation...exhibits"
The text has been corrected.

L28 "theatre" change to "location"
The word has been replaced.

L34 " and to the seasonal restratification" - this hasn't been shown in the paper. In any event, the seasonal restratification is likely to be driven primarily by heating at the surface. This is shown by the fact that a 1D model in the vertical produces adequate seasonal restratification without any MLIs.
This part has been rephrased taking into account referee remark.

L36 " Indeed, Soufflet et al (2016)" - this isn't included in the list of references.
The reference has been added.

A number of figures are missing x, y and colorbar axis labels. The text on the labels is also generally quite small and hard to read. Maybe it could be made into a semi-bold font?
We apologize but we do not understand this point. On our version of the manuscript, labels and associated text are clearly visible (with a resolution allowing to zoom without loosing information). We would be happy to reprocess any figures if the problem still appears in the revised manuscript.

Figure 1 - the non-uniform color scheme here is not a good choice - it strongly highlights the yellow region which may or may not be the region of strongest gradients. A uniform perception scheme is recommended instead.
The figure 1 has been modified following referee suggestion using a uniform perception colormap.

5 Figure 2 - the title should be in symbols rather than the computer code format used. This is confusing as it's not clear to me if the < > are being used as mathematical operators here or just part of your text formatting. The temporal averaging that is required to make the calculation in the plot should be made explicit in the text. "The shape of the curves represents the spatial standard deviation" - this seems incorrect, the shape of the curves is the seasonal cycles, but it seems like you're actually talking about the shading around the curves.
10 We agree with the referee. The title of the figure is confusing has been modified.
In the text, the sentence has been rephrased as:
" The shading around the curves represents the spatial standard deviation ..."

Figure 4 - why are there two subplots? Why is one of them smaller than the other? Does the second one even get referred to?
15 Figure 4 caption has been updated to describe both figures. The new caption is:
" Figure 4: Sea surface salinity (28th July 2009) during an event of freshwater export in the open ocean as described by Reverdin *et al.* (2013). Left figure is representing the full model domain and right figure is focused on the South-Eastern part of the Bay of Biscay to highlight the freshwater export around 44°N and 3.5°W."

20 Figure 6 - shouldn't these plots show the average over 50 grid points rather than just selecting one grid point out of 50?
We understand the referee suggestion. Before giving this figure, we tried different way of plotting this average circulation. In order to compare with Charria et al. (2013), this subsampling was making more sense than the average. A better solution would have been to average on the same grid as the one used in Charria et al. (2013) but it was not possible to do such computation in the review available time.
25

Figure 8 - a red-white-blue colorbar should be used instead of a rainbow color schemes. In general rainbow color schemes are known to distort perception of data and should be avoided. The figure titles could mention which is the model and which is the observation.
A blue-white-red colormap is used instead of the previous rainbow one. Observation data have been added to improve the
30 clarity of the manuscript instead of only referring to the corresponding publication.

In the manuscript the description of these figure has been updated and simplified following the new colormap and the observations:
" Other comparisons have been performed with an ADCP mooring during the ARCADINO experiment. This mooring
35 located on the Aquitaine shelf (Figure 1) has been used to highlight poleward coastal jets up to 32 cm s$^{-1}$ (Batifoulier et al., 2012). Similar events are modelled in our numerical experiments with smaller amplitudes (Figure 8). In 2008, a poleward along-shore current appears around the 15th August 2008 (Figure 8) in observations. From in situ ADCP measurements (Batifoulier et al., 2012), it also appears a velocity maximum between the 16th and 20th August 2008. In the modelled fields, the jet is reproduced but velocities are weaker and the event starts earlier in the simulation. The jet is also deeper in the
40 model (20-40 m depth with maximum velocities ~16 cm s$^{-1}$) than in observations with a maximum above 30 m depth. When model forcings are explored, we explain this event with similar conditions to those observed in Batifoulier et al. 2012. Indeed, westerly winds are blowing from the 6th to the 8th August (with intensities 8 to 12 m s$^{-1}$) along the Spanish coast to setup the circulation resulting in the poleward jet following the explained process in Batifoulier et al. (2012)."

45 Figure 9 - it is more helpful to plot the relative vorticity normalised by the local value of f (and other vorticity plots).
Figures have been modified following referee suggestion.

Figure 12 - the depth of the plan views needs to be stated.
In the figure caption, the depth is given (4-m depth).

---

## Author Comment (AC2) · 21 May 2017

This is a relatively short paper documenting the seasonal variability in a high resolution, regional numerical model of the upper ocean in the Bay of Biscay. It is first demonstrated that the model faithfully reproduces the large space and time scale vari- ations in the region. The authors then go on to describe the level of submesoscale variability through the spectral energy of vorticity and vertical buoyancy flux. A relationship between the depth of winter mixing and the energy in the submesoscale is observed.
This paper is well written and the model seems to provide a useful representation of the meso- to submesoscale variability in this region. However, aside from documenting the realism of this particular model, I do not find anything new or novel in the paper. Theories exists that relate submesoscale energy to mixed layer depth (among other things), so the present qualitative finding is entirely expected. It is stated that their results show the importance of submesoscale activity, but this is really only implied. It could be demonstrated by comparison with an otherwise identical model that did not resolve the submesoscale, but this is not done. First one would have to define what quantity they were interested in. It is possible that the submesoscale is important for some things and not for others. The title is somewhat misleading since the subme- soscale is discussed in only about 1 1/2 pages and the interannual in only one short paragraph.
I see this as an editorial decision. I did not really learn much from reading the paper, but it does fairly clearly document some aspects of the fidelity of this regional model. If that fits the goals of the journal then the paper could probably be suitable for publication with some revisions. However, my own recommendation would be to reject the paper since I do not see any reasonable revisions leading to new insights. That is not to say that the model does not contain new and interesting things that could be explored and understood, it is just that does not seem to be the authors objective for writing this paper.

Following major referee comments, the result/discussion parts have been modified including more emphasized results and discussion on the interannual variability. The relationship between forcing terms and ocean features has been discussed including new details on the process explained. However, this remains more limited than expected as we were facing a technical problem to give a full detailed description of the turbulent fluxes (latent and sensitive fluxes) as they have not been saved for this simulation and we do not have the available computing time and infrastructure to perform the same simulation with more saved variables. Nevertheless, we explored the different possible improvements based on offline diagnostics and we propose more detailed analyses to confirm our proposed assumptions in the discussion.

Compared with the previous manuscript, a discussion has also been added on the interannual evolution of the spectral content based on spectral slope analysis.

We believe that the results presented in this paper are leading to a new understanding of the interannual dynamics in the Bay of Biscay and the associated evolution of (sub)mesoscale.

Detailed comments:
(Page 2, line 8): There really isn't any connection in the paper between submesoscale activity and climate change.
Following referee suggestion, the text has bee modified as:
"... long-term fluctuations related to evolutions in atmospheric conditions. ..."

(3,23) More details are needed on the lateral boundary conditions. Is the sponge layer just a region of high viscosity or are the model prognostic variables are restored to- wards the ORCA12 variables? Are the ORCA12 variables interpolated and imposed on the boundaries? If so, are the tidal components then added? Just velocity or do the tides perturb the density field as well? Is anything done to sea surface height? What is the temporal resolution of the ORCA12 data?
An appendix has been added to the manuscript in order to clarify all those details.
Concerning the lateral boundary conditions, variables are interpolated and imposed on the boundaries to ORCA12. Lateral boundary conditions are prescribed in velocity, temperature, salinity and sea surface height.
The temporal resolution of ORCA12 data is 5 days.

(3,26) Is there no restoring for salinity, just a surface flux boundary condition?
There is no restoring for salinity.

(4,15) It would be helpful to indicate the annual mean values for the model and observations on the figure.
As the time series is giving the mean values for each day, we do not really understand what is the referee suggestion.

(4,21) It would be helpful here and elsewhere to mark on the figures various geographic features described in the text to help orient the reader.
Indeed, we agree with the referee and we added in the text a geographical position close to mentioned places not given on the figures.

More statistics have been given in the text. However, the dataset does not have enough profiles to conclude on the quality of the simulation to reproduce eddies.

We agree with the referee but after several tries, it appears as the best representation for these fields. More generally, fonts have been enlarged in the figures.

Indeed, for this comparison, we are checking the model/observation agreement, which is linked to both forced and internal variability.

As suggested by the referee, observations have been included here.

We agree that the topographic contour can be useful to see the transition. However, the red/blue colormap is not adapted to overplot isobaths. We then referred to Figure 1 and Figure 6 including the detailed view of isobaths.

This refers to another study focused on frontal activity over the shelf by O. Yelekci. We analysed this structures and related it to the local drivers (mainly rivers and winds). We add a mention to a personal communication.

This question is very interesting question that we cannot address with our realistic (expensive) simulation. Further simplified modelling are planned to explore these issues.

We agree with the referee, however, as we consider the vertical velocities on the first 150m-depth, we can assume that it will impact the vertical mixing.

We agree with the referee that it remains a seasonal signal in salinity related to the evaporation-precipitation effect. The text has been modified to mention it:
" As we consider an area not under direct influence of major river runoffs (far from the slope dynamic barrier), the salinity (Figure 14d) does not exhibit a regular seasonal cycle. Indeed, main sources of freshwater in the Bay of Biscay are coming for river discharges. These discharges follow a seasonal cycle with a maximum flow end of winter not simulated over the analysed domain. Furthermore, the evaporation-precipitation budget (related to the more intense and frequent depression in winter) does not induce large variations at seasonal scales in the region but fluctuates interannually depending the atmospheric conditions. "

The sentence has been rephrased, as it was not clear for the reader. The text in the manuscript has been modified as:
" These instabilities drive to a conversion in kinetic energy of the stored potential energy in winter and lead to reinforce the seasonal stratification."

In the new manuscript, we managed to strengthen the part and discussion on interannual variability as it remains the main targeted focus of the paper. However, we had to keep an important validation part as it is expected when we infer analyses on realistic simulations.

---

## Referee Report (RR1)

L 17 "Earth's rotation

p3 L17 "where h is the depth"?

L24 "width is 20 km"

L25 "The tide with 14..."

L33 "A spin-up of two years is required taking into account

P4 L11 - this equation (and others on this page) need to be on their own line

P5 L4 "better resolve"

L10-15 - this could all be deleted, readers can look up the database construction themselves if interested.

L39 "The Figure 6" should be "Figure 6" with no article (and elsewhere)

P6 L38 "there is also a velocity..."

P7 L15 "at different contrasted"

P8 L9 "vertical mixing" - I think you really mean vertical transport here as the discussion focuses on resolved vertical motions rather than parameterised vertical mixing.

L26 - missing a "x" here in the number perhaps?

L36 "and associated with the larger anomalies compared with the averaged annual cycle" - I don't understand what this means?

P9 1 - "are observed ... 8 year period"

L5 "from" "at the end"

L6 "depression" to "cyclonic weather systems"

L10 - the use of 'maximum' and 'small' is a bit confusing given the negative numbers. Perhaps 'steepest' and shallowest' could be used instead?

L11 - I'm confused about the November year stuff

L20 "low levels"

L34 "4 km"

P11 L4 "biogeo"

L13 "into the Bay..."

L15 "associated with" (and elsewhere)

L16 "also reach deep mixed layer depth maxima"

L20 "On the contrary,... had shallower mixed layer depth maxima ... reduced"

L21 ". These shallower mixed layers are related ..."

L23 "recorded"

L34 "autumn 2004"

P11 L14 "warm winters"

P12 L21 - delete comma

Appendix A - I'm not suggesting you need to do anything for this paper, but I'd recommend preparing papers in some variation of Latex to avoid the slightly strange appearance of the equations that happens with Word.

---

## Author Response (AR2)

L. Brannigan (Referee)
Submitted: 26 June 2017

L 17 "Earth's rotation

The manuscript has been modified following referee suggestion.

p3 L17 "where h is the depth"?

The text has been corrected.

L24 "width is 20 km"

The manuscript has been corrected.

L25 "The tide with 14..."

The sentence has been modified.

L33 "A spin-up of two years is required taking into account

The sentence has been improved following suggestion.

P4 L11 - this equation (and others on this page) need to be on their own line

Equations of p.4 have been written on their own lines.

P5 L4 "better resolve"

The text has been corrected.

L10-15 - this could all be deleted, readers can look up the database construction themselves if interested.

Following referee suggestion, this part has been simplified.

L39 "The Figure 6" should be "Figure 6" with no article (and elsewhere)

The text has been modified.

P6 L38 "there is also a velocity..."

The manuscript has been rephrased following suggestion.

P7 L15 "at different contrasted"

The text is following referee suggestion.

P8 L9 "vertical mixing" - I think you really mean vertical transport here as the discussion focuses on resolved vertical motions rather than parameterised vertical mixing.

Indeed, we agree with the referee and the text has been modified. "vertical mixing" => "these vertical motions"

L26 - missing a "x" here in the number perhaps?

As it depends the journal conventions, we followed the referee suggestion and add the 'x'.

L36 "and associated with the larger anomalies compared with the averaged annual cycle" - I don't understand what this means?

The sentence has been replaced to improve the reader understanding of this result:

" The maxima are in phase with the coldest period in temperature and associated with the larger anomalies compared with the averaged annual cycle (Figure 14c) before the spring warming and the beginning of seasonal stratification."

has been replaced by

"The maxima are in phase with the coldest period in temperature and most extreme values in vertical velocity and relative vorticity are corresponding to the most extreme cold values in temperature compared with the annual cycle (Figure 14c) before the spring warming and the beginning of seasonal stratification."

P9 1 - "are observed ... 8 year period"

The manuscript has been corrected.

L5 "from" "at the end"

The manuscript has been corrected.

L6 "depression" to "cyclonic weather systems"

The sentence has been modified.

L10 - the use of 'maximum' and 'small' is a bit confusing given the negative numbers. Perhaps 'steepest' and shallowest' could be used instead?

Indeed, the use of this term is not the most suitable. We modified the text using "steep" and "shallowest" terms. The manuscript text is then:

"The Figure 15 shows the shallowest slopes (larger than $k^{-0.4}$) occurring in autumn/winter (from November to March). At the opposite, slopes values are steeper (between $k^{-1.2}$ and $k^{-1.4}$) in spring with a minimum in May or June. The interannual variability of this minimum (corresponding to steepest slopes) is limited and values are very similar following the year. Concerning the winter shallowest slopes, the value is decreasing with time but the limited number of simulated years does not allow concluding to the significance of this trend. The monthly seasonal cycle is very stable every year. However, we can notice that in 2004, shallowest slopes are reached earlier (in November) than during the other years (December, January or February)."

L11 - I'm confused about the November year stuff

To clarify the text, we replaced "... in winter (from November - year-1 to March year) ..." by "... in autumn/winter (from November to March) ...".

L20 "low levels"

The text has been corrected.

L34 "4 km"

The text has been corrected.

P11 L4 "biogeo"

The text has been corrected.

L13 "into the Bay..."

The manuscript has been modified.

L15 "associated with" (and elsewhere)

All occurrences of "associated to" have been replaced by "associated with".

L16 "also reach deep mixed layer depth maxima"

The manuscript has been modified.

L20 "On the contrary,... had shallower mixed layer depth maxima ... reduced" L21 ". These shallower mixed layers are related ..."

The text has been adapted to take the suggestion into account.
Improved manuscript text is:
" On the contrary, 2007 and 2008 had shallower mixed layer depth maxima (Figure 16a) associated with a reduced maximum of vertical buoyancy flux at (sub)mesoscale (Figure 16b). These shallower mixed layers are related to warm winters causing warming of the surface ocean and a decrease in winter mixing."

L23 "recorded"

The text has been corrected.

L34 "autumn 2004"

The text has been corrected.

P11 L14 "warm winters"

The text has been corrected.

P12 L21 - delete comma

The text has been corrected.

Appendix A - I'm not suggesting you need to do anything for this paper,
but I'd recommend preparing papers in some variation of Latex to avoid the slightly strange appearance of the equations that happens with Word.

We thank the referee for these future recommendations.

---

## Author Response (AR3)

**Reply to* editor comment on "Interannual evolutions of (sub)mesoscale dynamics in the Bay of Biscay" by Guillaume Charria et al.**

**Topic editor**
Submitted: 3 August 2017

p8, l37: "most extreme" --> "the most extreme"
p9, l12: "The Figure 15" --> "Figure 15"
p9, l19: "spatial scales distribution" --> "the distribution of spatial scales"

The manuscript has been modified following editor suggestions.